# Cross-Device Collaborative Test-Time Adaptation

**Guohao Chen**[1 2*]  **Shuaicheng Niu**[3*]  **Deyu Chen**[1]  **Shuhai Zhang**[1 2]
**Changsheng Li**[4†]  **Yuanqing Li**[1 2]  **Mingkui Tan**[1 2†]
[1]South China University of Technology, [2]Pazhou Laboratory,
[3]Nanyang Technological University, [4]Beijing Institute of Technology,

## Abstract

In this paper, we propose test-time **Co**llaborative **L**ifelong **A**daptation (CoLA), which is a general paradigm that can be incorporated with existing advanced TTA methods to boost the adaptation performance and efficiency in a multi-device collaborative manner. Specifically, we maintain and store a set of device-shared *domain knowledge vectors*, which accumulates the knowledge learned from all devices during their lifelong adaptation process. Based on this, CoLA conducts two collaboration strategies for devices with different computational resources and latency demands. 1) Knowledge reprogramming learning strategy jointly learns new domain-specific model parameters and a reweighting term to reprogram existing shared domain knowledge vectors, termed adaptation on *principal agents*. 2) Similarity-based knowledge aggregation strategy solely aggregates the knowledge stored in shared domain vectors according to domain similarities in an optimization-free manner, termed adaptation on *follower agents*. Experiments verify that CoLA is simple but effective, which boosts the efficiency of TTA and demonstrates remarkable superiority in collaborative, lifelong, and single-domain TTA scenarios, *e.g.*, on follower agents, we enhance accuracy by over 30% on ImageNet-C while maintaining nearly the same efficiency as standard inference. The source code is available at https://github.com/Cascol-Chen/COLA.

## 1 Introduction

The conventional pipeline of deep learning typically trains a model and deploys it across numerous devices with frozen parameters. This pipeline has demonstrated great success in various applications, such as autonomous driving cars [35, 14], embodied robots [18], and many other smart devices [28, 29]. However, during deployment, the model on each device may encounter test samples drawn from a domain different from the training one. In some cases, the testing environment even changes continuously and periodically, such as changes in weather. Unfortunately, the deep model often struggles to generalize to unseen testing domains and its performance may degrade significantly.

To resolve domain shifts, test-time adaptation (TTA) [51, 21, 54, 64, 2, 7, 59, 26, 39, 25, 52, 37] has emerged as a promising research field. TTA updates a given model w.r.t. a testing sample using self-/unsupervised objectives, such as rotation prediction [13], contrastive learning [2, 30, 50], entropy minimization [54, 64, 25, 37], *etc*. Compared to conventional domain adaptation [31, 45, 24] or fine-tuning [61, 33] methods that require performing offline model learning on the whole pre-collected target dataset, TTA distinguishes itself with minimal overhead by utilizing each test sample only once for immediate post-inference adaptation. This renders TTA more adaptable in real-world applications.

However, prior TTA methods mainly validate their effectiveness on a single device, *i.e.*, re-adapting the model from scratch on each. In practice, models are often deployed across multiple devices.

---

*Equal contribution. Email: secasper@mail.scut.edu.cn, shuaicheng.niu@ntu.edu.sg
†Corresponding author. Email: mingkuitan@scut.edu.cn, lcs@bit.edu.cn

38th Conference on Neural Information Processing Systems (NeurIPS 2024).

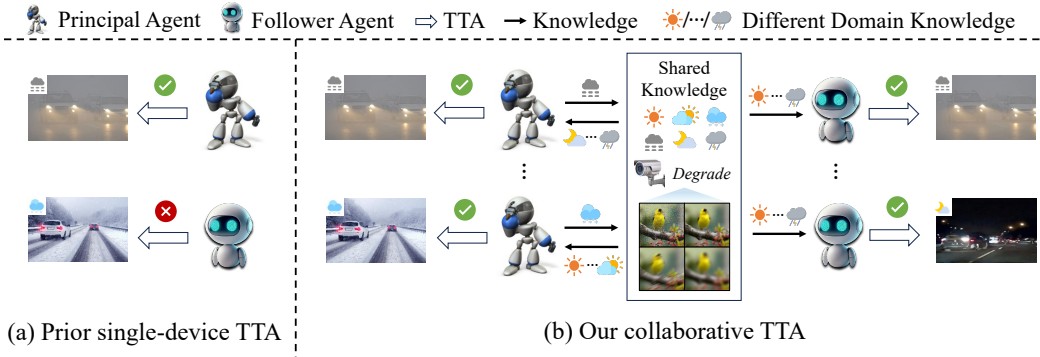

🧑‍🔬 Principal Agent  🤖 Follower Agent  ⇨ TTA  → Knowledge  ☀️/⋯/🌧️ Different Domain Knowledge

(a) Prior single-device TTA  (b) Our collaborative TTA

Figure 1: Comparison w.r.t. (a) prior single-device TTA *vs.* (b) our collaborative TTA. Prior TTA operates on each device independently and may be infeasible in resource-limited devices. In contrast, our collaborative TTA allows devices to share knowledge. Based on this, on different devices, one can choose to solely aggregate the shared knowledge for TTA (*Follower Agents*), or further conduct backpropagation for knowledge aggregation and new domain knowledge learning (*Principal Agents*).

As shown in Figure 1, in the multi-device adaptation scenario, single-device adaptation methods suffer from the following limitations. **First**, single-device TTA neglects useful knowledge learned from other devices and adapts independently. Since different devices may frequently encounter similar or even identical testing domains, ignoring this shared knowledge often leads to suboptimal adaptation performance, as demonstrated in Table 2. **Second**, due to limited resources or latency demands, some devices may not support the backpropagation operation required by learning-based TTA methods [54, 38], rendering single-device TTA infeasible. **Third**, even on a single device, models may also encounter dynamically and periodically changing domain shifts. Although recent works have proposed continual TTA to mitigate the catastrophic forgetting issue, such as anti-forgetting regularizer [38] or restoration schemes [56], these methods still struggle to accumulate previously learned knowledge over a long-term adaptation process, as shown in Table 1.

In this paper, we propose a test-time **Co**llaborative **L**ifelong **A**daptation (CoLA) method to enable knowledge accumulation, sharing, and exploitation across devices. Our approach exploits both the previously learned knowledge from other devices and the device itself to achieve efficient and collaborative TTA. Specifically, we represent the knowledge learned on each domain of each device by a domain vector, and automatically detect domain changes on each device during its continual adaptation process. These domain vectors are stored and shared across devices upon domain changes for collaborative TTA and catastrophic forgetting mitigation. Based on the shared domain vectors, we first introduce a knowledge reprogramming learning method for *Principal Agents*, *e.g.*, the resource-abundant devices, where we enhance TTA performance and efficiency by leveraging available shared knowledge while learning new domain-specific parameters in case existing knowledge is insufficient. The newly learned parameters/knowledge are subsequently stored for shared domain vector set updating. Furthermore, we devise an optimization-free collaborative TTA method, to reduce the computation consumption of TTA and thus enable TTA in latency-sensitive scenarios or resource-limited devices, which we term as *Follower Agents* in CoLA. We achieve this by directly aggregating the domain knowledge shared by principal agents according to domain similarities.

**Main novelty and contributions.** 1) We introduce a novel and practical collaborative lifelong adaptation paradigm to TTA. This paradigm addresses a practical demand in real-world applications to perform effective adaptation on numerous devices with varying resources and latency requirements simultaneously, meanwhile keeping privacy preserved and communication efficient. 2) We devise domain vectors to explicitly store the domain knowledge and share them across devices for collaborative TTA. Based on this, we devise two collaborative strategies, *i.e.*, knowledge reprogramming learning for resource-abundant principal agents and similarity-based knowledge aggregation for resource-limited or latency-sensitive follower agents. 3) Extensive experiments demonstrate the superiority of our CoLA regarding the scenarios of collaborative, lifelong, and single-domain TTA in a plug-and-play manner. By leveraging available shared knowledge, on principal agents, we achieve an up to 78.0× speed up on ETA compared with the baseline without collaborative learning on ImageNet-C. On follower agents, we enhance the accuracy by over 30% while maintaining nearly the same computation and memory efficiency as standard inference on ImageNet-C.

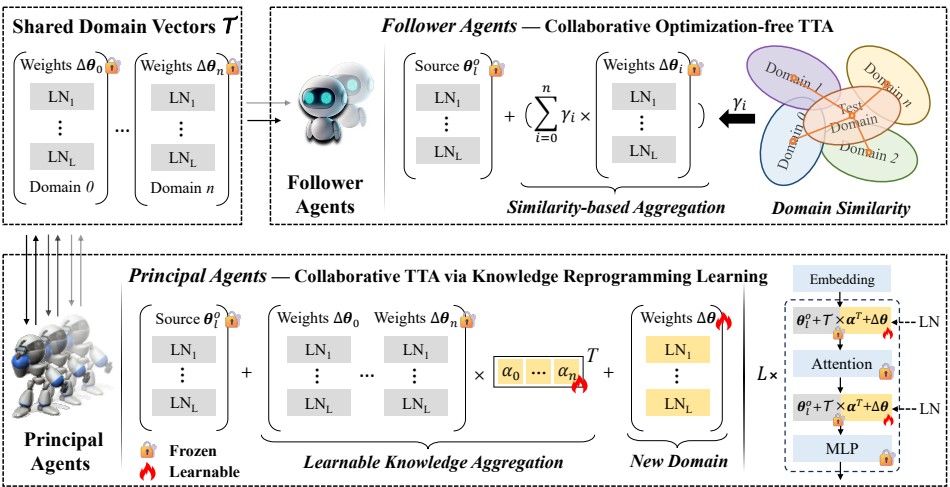

Figure 2: An illustration of our proposed CoLA. We maintain a shared domain vector set $\mathcal{T}$ to explicitly store the knowledge learned by each principal agent during adaptation. Based on $\mathcal{T}$, for *Principal Agents*, we jointly learn the domain-specific parameters $\Delta\boldsymbol{\theta}$ and the reweighting term $\boldsymbol{\alpha}$ via backward propagation, where the learned knowledge is then stored in $\mathcal{T}$. For *Follower Agents*, we adaptively aggregate the shared knowledge in $\mathcal{T}$ in a forward-only manner, based on the domain similarities, which prioritizes knowledge derived from domains that are similar to the testing domain.

## 2  Problem statement and motivation

Let $f_{\boldsymbol{\theta}}(\cdot)$ be the model trained on a labeled dataset $\mathcal{D}_{train} = \{(\mathbf{x}_i, y_i)\}$ and $\mathbf{x}_i \sim P(\mathbf{x})$. After training, $f_{\boldsymbol{\theta}}(\cdot)$ is often deployed on various devices, on each device $f_{\boldsymbol{\theta}}(\cdot)$ may encounter test samples drawn from a shifted and dynamically changing domain distribution $Q(\mathbf{x})$, where $Q(\mathbf{x}) \neq P(\mathbf{x})$. Under such domain shifts, deep models are often very sensitive and suffer from severe performance degradation. To address this, on each device, one can adapt $f_{\boldsymbol{\theta}}(\cdot)$ to $\mathbf{x}$ by optimizing some self-/unsupervised learning objective at test time:

$$\min_{\boldsymbol{\theta}_l} \mathcal{L}(\mathbf{x}; \boldsymbol{\theta}_f, \boldsymbol{\theta}_l), \quad \mathbf{x} \sim Q(\mathbf{x}), \tag{1}$$

where $\boldsymbol{\theta}_f$ and $\boldsymbol{\theta}_l$ denote frozen and learnable model parameters, respectively.

**Motivation.** Eqn. (1) is known as a single-device test-time adaptation (TTA) method, which naively re-adapts $f_{\theta}(\cdot)$ from scratch on each device. In our multi-device adaptation scenario, this independent adaptation manner neglects the valuable knowledge learned from other devices and often obtains limited performance, as in Figure 3(a). Therefore, there is an urgent demand to devise multi-device collaborative TTA methods, to enhance the adaptation performance and efficiency. To this end, the key technical challenge lies in devising a collaboration scheme that effectively exploits the knowledge from other devices while ensuring privacy preservation and communication efficiency.

**Domain knowledge vectors.** In real-world scenarios, a model is often deployed in environments that may change continuously and cyclically, *e.g.*, day $\rightarrow$ night $\rightarrow$ day. Moreover, the model deployed on different devices may encounter similar environments, experiencing similar domain shifts. In such cases, when a device encounters test samples drawn from a domain previously seen by itself or by other devices, it is unnecessary to conduct adaptation from scratch. Instead, leveraging the previously acquired knowledge can achieve enhanced adaptation. Inspired by this, we seek to explicitly store the knowledge learned on each domain of each device, and then exploit this knowledge for collaborative TTA. We term this knowledge as domain vectors and introduce its definition below.

Formally, given $m$ devices with each having $n$ domains, we use a domain vector $\Delta\boldsymbol{\theta}^{i,j} = \boldsymbol{\theta}_l^{i,j} - \boldsymbol{\theta}_l^o$ to denote the knowledge learned on the $i$-th domain of the $j$-th device. Here, $\boldsymbol{\theta}_l^{i,j}$ denotes the learned parameters on the $i$-th domain of the $j$-th device, and $\boldsymbol{\theta}_l^o$ denotes the corresponding original learnable parameters. For privacy and efficiency considerations, we select the **affine parameters of the norm layers** as learnable parameters $\boldsymbol{\theta}_l$ and transmit domain vectors between devices for knowledge sharing, which consumes negligible extra cost as in Table 5. We store each knowledge vector $\Delta\boldsymbol{\theta}^{i,j}$ in a set

**Algorithm 1** The overall pipeline of CoLA.

---

**Input:** Test samples $\{\mathcal{D}^m\}_{m=1}^M$, where $\mathcal{D}^m=\{\mathbf{x}_t^m\}_{t=1}^T$ denotes the batches of test samples on the $m$-th device, the model $f_{\boldsymbol{\theta}}(\cdot)$ and its stem (first) layer $f_{\boldsymbol{\theta}_s}(\cdot)$, threshold $z$.
 1: Initialize: shared domain vectors $\mathcal{T}=\{\mathbf{0}\}$, $\phi_d=\mathbf{0}$ for each device.
 2: **for** $t=1,2,\ldots,T$ **do**
 3:   **for** each device **(in parallel) do**
 4:     Calculate batch statistics $\hat{\phi}_t$ over $f_{\boldsymbol{\theta}_s}(\mathbf{x}_t^m)$;
 5:     Update distribution statistics $\phi_d$ by Eqn. (6);
 6:     `For Principal Agent:`               *// knowledge reprogramming learning, Sect. 3.1*
 7:       **if** $D(\phi_d, \hat{\phi}_t) > z$ **then**                     ▷ *domain change detection, Eqn. (7)*
 8:         Update domain vectors $\mathcal{T}$ by storing the newly learned $\Delta\boldsymbol{\theta}$ and reset $\phi_d=\mathbf{0}$.
 9:       **end if**
10:       Predict $\hat{y}_t^m$ by $f_{\boldsymbol{\theta}}(\mathbf{x}_t^m)$ based on Eqn. (2);
11:       Update $\boldsymbol{\alpha}$ and $\Delta\boldsymbol{\theta}$ via Eqn. (2) with backpropagation;
12:     `For Follower Agent:`               *// similarity-based aggregation, Sect. 3.2*
13:       Update $\rho_i$ for different domain knowledge via Eqn. (5);
14:       Predict $\hat{y}_t^m$ by $f_{\boldsymbol{\theta}}(\mathbf{x}_t^m)$ based on Eqn. (3);
15:   **end for**
16: **end for**
**Output:** Predictions $\{\hat{y}_t^m \mid t=1,...,T \text{ and } m=1,...,M\}$.

---

$\mathcal{T}=\{\Delta\boldsymbol{\theta}^{i,j}\}_{i=1,j=1}^{n,m}$. During the continual adaptation process, we dynamically expand $\mathcal{T}$ by storing a new $\Delta\boldsymbol{\theta}^{i,j}$ in $\mathcal{T}$ once $\Delta\boldsymbol{\theta}^{i,j}$ is learned. For simplicity of presentation, we omit $\mathcal{T}$ as $\{\Delta\boldsymbol{\theta}_i\}_{i=1}^N$ where $N=mn$ and exploit $\mathcal{T}$ to devise collaborative TTA strategies in the following sections.

## 3 Cross-device collaborative test-time adaptation

In this paper, we propose a test-time **Co**llaborative **L**ifelong **A**daptation (CoLA) method. In CoLA, we seek to conduct collaborative TTA across multiple devices by exploiting a set of shared **domain vectors**, termed $\mathcal{T}$. We automatically detect the domain changes on each device during a continual adaptation process and explicitly store the current domain knowledge in $\mathcal{T}$ once the testing domain is changed (c.f. Sect. 3.3). Then, based on $\mathcal{T}$, we develop two distinct collaboration strategies. In practice, users can determine which strategy to use according to the computational resource of their device or their latency requirements. **First**, the collaborative knowledge reprogramming learning strategy (c.f. Sect. 3.1) is designed for "**principal agents**", *i.e.*, the devices that will dominate the learning of new knowledge and have sufficient resources for backpropagation-based model updates. This strategy jointly learns new domain-specific model parameters and a reweighting term to reprogram the knowledge learned from previously encountered distributions from both the device itself and other devices, through backpropagation-based optimization. **Second**, the optimization-free collaborative TTA (c.f. Sect. 3.2) is designed for "**follower agents**", *i.e.*, the devices that are resource-limited or latency-sensitive. This strategy mainly exploits the knowledge shared from principal agents, by aggregating the valuable shared knowledge according to distribution similarities. We summarize the pseudo-code in Algorithm 1 and illustrate the overall pipeline of CoLA in Figure 2.

### 3.1 Collaborative test-time adaptation via knowledge reprograming learning

We conduct collaborative adaptation with both the knowledge learned from other devices and the device itself. This latter one is particularly advantageous for lifelong adaptation since, in practice, a deployed model may encounter diverse and evolving domains. By storing and reprogramming all knowledge learned from previously encountered domains, we naturally mitigate the issue of catastrophic forgetting during lifelong TTA (see results and analysis in Table 1). To this end, on each device, we detect domain changes and store the learned model parameters for each specific domain once the test domain changes. We then reprogram the knowledge stored in these model parameters by reweighting as below, facilitating seamless adaptation to changing environments.

As aforementioned, we assume there exist $N$ sets of parameters, *i.e.*, domain vectors, learned from previously encountered domains across all devices, denoted as $\mathcal{T} = \{\Delta\boldsymbol{\theta}_i\}_{i=1}^N$, where $\Delta\boldsymbol{\theta}_i = \boldsymbol{\theta}_i - \boldsymbol{\theta}_l^o$. For flexibility, we denote null knowledge as $\Delta\boldsymbol{\theta}_0 = \mathbf{0}$ and include it in $\mathcal{T}$. We use $\mathcal{T}$ to illustrate our collaborative learning scheme here and put details for detecting and storing each $\Delta\boldsymbol{\theta}_i$ in **Sect. 3.3**. Based on $\mathcal{T}$, we learn a reweighting term that adaptively aggregates shared knowledge via backpropagation, while learning new knowledge simultaneously if existing knowledge is insufficient. The overall optimization problem is given by:

$$\min_{\boldsymbol{\alpha}, \Delta\boldsymbol{\theta}} \mathcal{L}(\mathbf{x}; \boldsymbol{\theta}_f, \boldsymbol{\theta}_l), \ \ \text{where} \ \boldsymbol{\theta}_l = \boldsymbol{\theta}_l^o + \sum_{i=0}^N \alpha_i \Delta\boldsymbol{\theta}_i + \Delta\boldsymbol{\theta} \ \text{ and } \Delta\boldsymbol{\theta}_i \in \mathcal{T} = \{\Delta\boldsymbol{\theta}_i\}_{i=0}^N. \quad (2)$$

Here, $\Delta\boldsymbol{\theta}$ denotes learnable parameters for current round of adaptation, and $\boldsymbol{\alpha}$ denotes the normalized weights for different domain knowledge, *i.e.*, $\sum_i \alpha_i = 1$. Thus, knowledge reprogramming and new knowledge learning are decoupled into the optimization of $\boldsymbol{\alpha}$ and $\Delta\boldsymbol{\theta}$. Note that we introduce $\Delta\boldsymbol{\theta}_0$ to ensure more flexibility in reprogramming, *e.g.*, by setting $\alpha_0 = 1$, one can entirely disregard previously learned knowledge when it is not beneficial for adapting to currently encountered domain. Moreover, when no knowledge has been accumulated, *i.e.*, $\mathcal{T} = \{\mathbf{0}\}$, Eqn. (2) simplifies to the conventional TTA. We put details of the initialization for $\boldsymbol{\alpha}$ and $\mathcal{T}$ in Appendix B due to page limits.

**Adaptive temperature scaling for fast adaptation.** Promptly re-weighting the appropriate knowledge for aggregation is key to fast adaptation when the testing distribution suddenly changes. However, this process is hindered when the logits of $\boldsymbol{\alpha}$ are sharpened, making re-weighting difficult to favor another knowledge. To mitigate this, we further introduce an adaptive scaling temperature during the optimization phase, which helps adjust the sharpness of $\boldsymbol{\alpha}$ adaptively while maintaining the smoothness of the original logits. The calculation of $\boldsymbol{\alpha}$ can be thus expressed as $\boldsymbol{\alpha} = \mathrm{softmax}(\boldsymbol{\beta} \cdot T_l)$, where $\boldsymbol{\beta}$ is the logit vector and $T_l$ is a learnable temperature.

## 3.2 Collaborative test-time adaptation via similarity-based knowledge aggregation

The computational constraints of devices, combined with the real-time demands of various applications, often necessitate TTA to be as efficient as possible. To this end, we propose a forward-only collaborative TTA strategy for devices operating in the follower agent mode. Here, the follower agent adapts a given model by aggregating the previously learned knowledge and the knowledge shared from principal agents without learning new domain-specific parameters, aiming to maintain almost the same efficiency as pure inference. Formally, given domain vectors $\mathcal{T} = \{\Delta\boldsymbol{\theta}_i\}_{i=0}^N$ shared from $m$ principal resource-abundant devices with each having $n$ encountered domains and $\Delta\boldsymbol{\theta}_0 = \mathbf{0}$, the goal of adaptation is to find appropriate normalized weights $\boldsymbol{\gamma}^* \in \mathbb{R}^{N+1}$ such that:

$$\boldsymbol{\gamma}^* = \arg\min_{\boldsymbol{\gamma}} \mathcal{L}(\mathbf{x}; \boldsymbol{\theta}_f, \boldsymbol{\theta}_l), \ \ \text{where} \ \boldsymbol{\theta}_l = \boldsymbol{\theta}_l^o + \sum_{i=0}^N \gamma_i \Delta\boldsymbol{\theta}_i \ \text{ and } \Delta\boldsymbol{\theta}_i \in \mathcal{T} = \{\Delta\boldsymbol{\theta}_i\}_{i=0}^N. \quad (3)$$

Here, since backpropagation-based learning is not supported, we directly assign the specific value of each $\gamma_i$ approximately according to distribution similarities. We estimate the distribution by calculating the feature statistics, *i.e.*, the mean and standard deviation of the features from the first stem layer. Formally, let $\phi_d$ denote the statistics of the current distribution that is online estimated via Eqn. (6), and $\phi_i$ be the distribution statistics on which $\Delta\boldsymbol{\theta}_i$ is adapted (*i.e.*, the corresponding $\phi_d$ while learning $\Delta\boldsymbol{\theta}_i$), we re-weight the knowledge from different distributions by:

$$\boldsymbol{\gamma} = \mathrm{softmax}(\boldsymbol{\rho}), \quad \text{where} \ \rho_i = 1 \, / \, \big(D(\phi_d, \phi_i) + \epsilon\big). \quad (4)$$

Here, $\boldsymbol{\rho}$ is a logit vector, $D(\cdot, \cdot)$ is a distance for which we adopt KL divergence as in Eqn. (7), and $\epsilon$ is a small constant for numerical stability. In this way, we can adaptively prioritize shared knowledge learned from distributions that are similar to the current distribution. Note that a principal agent with abundant resources may also leverage Eqn. (3) for efficient TTA in real-time scenarios.

**Exploiting diverse knowledge for aggregation.** Aggregating the advantages of different knowledge is the key to achieving satisfying robustness under various distribution shifts, as shown in Figure 3 (b). However, when distributions are highly similar, the re-weighting logit $\rho_i$ shall become sufficiently large (*e.g.*, $\rho_i > 10$) and the softmax function tends to simplify to the max function, which hinders the potential of Eqn. (3) from aggregating a diverse set of knowledge. To alleviate this, we further introduce a pre-defined temperature scaling factor $T_f$ to soften $\rho_i$, thereby encouraging the aggregation

of more existing knowledge. Then, $\rho_i$ is re-defined as:

$$\rho_i = 1 \, / \, \big(T_f \cdot D(\phi_d, \phi_i) + \epsilon\big). \tag{5}$$

**Remark.** It is worth noting that our CoLA can be incorporated with existing TTA techniques as a plug-and-play module for a more effective solution (as in Table 1 and Table 3). Furthermore, unlike prior methods [4] that impose intensive transmission of both data and model weights, CoLA offers several merits for real-world implementation: 1) CoLA involves only the transmission of learned parameters $\Delta\boldsymbol{\theta}_i$, which preserves user privacy and imposes much less communication burden (*e.g.*, the affine parameters of the norm layers in ViT-Base [8], which are typically updated in TTA [54, 38], occupy only 0.15 MB). 2) the domain vectors are preserved and shared intermittently with a shift detector, which further reduces the communication burden by a considerable margin. 3) CoLA is decentralized and flexible, which allows all agents to join or leave the collaboration at any time.

### 3.3 Automatic domain shift detection for constructing domain knowledge vectors $\mathcal{T}$

In this section, we introduce the construction of the domain knowledge vectors $\mathcal{T}$ shared across multiple devices. As aforementioned, explicitly accumulating/storing the learned knowledge from each domain of each device in $\mathcal{T}$ is a key step in our CoLA for collaborative learning. However, in practice, during the lifelong adaptation process of each device, we do not have any prior information on the domain labels regarding a given test sample stream. To conquer this, we devise an efficient distribution shift detector to identify whether the test distribution changes, and then automatically store the currently learned model weights to the domain vector set $\mathcal{T}$ once the domain change is detected. We achieve this by measuring the discrepancy between the distribution's statistics $\phi_d$ and the statistics of the current test batch $\hat{\phi}_t$. Formally, let $f_{\boldsymbol{\theta}_s}(\cdot)$ be the stem layer of $f_{\boldsymbol{\theta}}(\cdot)$, *i.e.*, the first layer. $\hat{\phi}_t$ comprises the mean $\hat{\mu}_t$ and standard deviation $\hat{\sigma}_t$ calculated over $f_{\boldsymbol{\theta}_s}(\mathbf{x}_t)$. Then, we estimate the statistics $\phi_d$ from the observed test samples $\{\mathbf{x}_t\}_{t=1}^T$ via exponential moving average:

$$\phi_d = \lambda\hat{\phi}_t + (1-\lambda)\phi_d, \tag{6}$$

where $\lambda$ is a moving average factor belongs to $[0, 1]$. Inspired by existing distance-based detection methods [19], we capture the magnitude of distribution shifts by a distance function $D(\cdot, \cdot)$ as follows.

$$D(\phi_d, \hat{\phi}_t) = \frac{1}{H}\sum_{i=1}^{H} KL(\phi_{d,i}||\hat{\phi}_{t,i}) + KL(\hat{\phi}_{t,i}||\phi_{d,i}), \ \ KL(\phi_1||\phi_2) = \frac{1}{2\sigma_2^2}(\sigma_1^2 + (\mu_1 - \mu_2)^2), \tag{7}$$

where $H$ denotes the dimension of statistics, and $KL(\cdot||\cdot)$ is the KL-divergence simplified from [19]. Here, a distribution shift is detected when $D(\phi_d, \hat{\phi}_t) > z$, where $z$ is a pre-defined threshold. This simple design offers several merits: 1) It imposes minimal computational and memory costs without necessitating data preservation. 2) By leveraging the features from the stem layer, we can promptly detect and respond to distribution shifts, rendering it well-suited for the online nature of TTA.

## 4 Experiments

**Datasets and models.** We conduct experiments on the ImageNet-1k [6], as well as five benchmarks for OOD generalization, *i.e.*, ImageNet-C [16] (contains corrupted images in 15 types of 4 main categories and each type has 5 severity levels), ImageNet-R (various artistic renditions of 200 ImageNet classes) [15], ImageNet-Sketch [55], ImageNet-A [17], and ImageNet-V2 [44]. We use ViT-Base [8] as the source model unless stated otherwise. The model is trained on the source ImageNet-1K [6] training set and the model weights are obtained from the `timm` repository [60].

**Compared methods and implementation details.** We compared our proposed CoLA with 1) Backpropagation-based methods: CoTTA [56], ETA [38], EATA [38], SAR [39], and DeYO [25]. 2) Backpropagation-free methods: LAME [3] and T3A [21]. For Eqn. (2), We directly leverage the learnable test-time objectives from the integrated TTA methods. $\Delta\boldsymbol{\theta}$ is optimized by following the update rules of the integrated baseline as listed in Appendix C. $\boldsymbol{\alpha}$ is updated via the AdamW optimizer with a learning rate of 0.1. The shift detection threshold $z$ is set to 0.1. For follower agents, we consistently set $T_f$ in Eqn. (3) to 5 for all experiments. More details are put in Appendix A and C.

Table 1: Comparison on ImageNet-C (level 5) regarding **Accuracy (%)** under lifelong adaptation for 10 rounds, in total of 150 corruptions, on **a single principal resource-abundant device**. We report the average accuracy of 15 corruptions in each round here and put more results in Appendix D.

| Time: Round | 1 | 2 | 3 | 4 | 5 | 6 | 7 | 8 | 9 | 10 | Average |
|---|---|---|---|---|---|---|---|---|---|---|---|
| NoAdapt | 29.9 | 29.9 | 29.9 | 29.9 | 29.9 | 29.9 | 29.9 | 29.9 | 29.9 | 29.9 | 29.9 |
| CoTTA [56] | 44.9 | 40.6 | 35.8 | 32.7 | 30.4 | 28.9 | 27.7 | 27.1 | 27.2 | 26.5 | 32.2 |
| EATA [38] | 60.4 | 60.0 | 59.6 | 59.4 | 59.3 | 59.1 | 59.0 | 58.8 | 58.7 | 58.6 | 59.3 |
| SAR [39] | 59.1 | 60.6 | 60.9 | 61.2 | 61.3 | 61.4 | 58.3 | 60.4 | 60.8 | 61.1 | 60.5 |
| + CoLA (Ours) | 59.1 | 62.4 | 63.6 | 64.3 | 64.7 | 64.9 | 65.2 | 65.1 | 65.3 | 65.4 | $64.0_{(+3.5)}$ |
| ETA [38] | 61.4 | 58.7 | 54.5 | 50.2 | 46.2 | 44.1 | 38.8 | 38.0 | 36.7 | 35.1 | 46.4 |
| + CoLA (Ours) | 62.0 | 63.9 | 64.8 | 65.1 | 65.3 | 65.3 | 65.3 | 65.3 | 65.4 | 65.4 | $\mathbf{64.8}_{(+18.4)}$ |
| DeYO [25] | 59.8 | 48.8 | 0.1 | 0.1 | 0.1 | 0.1 | 0.1 | 0.1 | 0.1 | 0.1 | 10.9 |
| + CoLA (Ours) | 61.7 | 62.5 | 63.6 | 64.5 | 65.0 | 65.1 | 65.3 | 65.5 | 65.5 | 65.5 | $64.4_{(+53.5)}$ |

Table 2: Effectiveness under collaborative adaptation across **principal resource-abundant devices** w.r.t. **Accuracy (%)**. Results are evaluated on ImageNet-C (level 5, containing 15 corruption types of 4 groups). We share learned weights across devices post-adaptation to each group of corruptions.

| Method | Device 1 (Adapt →) | | | | Device 2 (Adapt →) | | | | Device 3 (Adapt →) | | | | Avg. |
|---|---|---|---|---|---|---|---|---|---|---|---|---|---|
| | Noise | Blur | Weat. | Digit. | Blur | Noise | Digit. | Weat. | Weat. | Digit. | Blur | Noise | |
| NoAdapt | 8.2 | 28.4 | 36.1 | 41.7 | 28.4 | 8.2 | 41.7 | 36.1 | 36.1 | 41.7 | 28.4 | 8.2 | 28.6 |
| CoTTA [56] | 28.9 | 41.3 | 50.2 | 47.6 | 36.3 | 37.1 | 50.4 | 52.7 | 50.4 | 55.0 | 42.5 | 38.7 | 44.2 |
| EATA [38] | 53.5 | 57.0 | 68.1 | 67.2 | 58.1 | 52.2 | 67.0 | 68.5 | 69.4 | 67.7 | 57.9 | 51.5 | 61.5 |
| SAR [39] | 50.4 | 54.4 | 66.3 | 64.5 | 55.1 | 48.3 | 64.0 | 66.4 | 66.5 | 64.3 | 55.4 | 47.7 | 58.6 |
| + CoLA (Ours) | 50.4 | 58.0 | 69.4 | 68.7 | 55.0 | 55.0 | 67.1 | 70.5 | 66.3 | 65.3 | 58.8 | 55.5 | 61.7 |
| ETA [38] | 55.2 | 56.9 | 67.5 | 66.0 | 59.8 | 51.7 | 65.0 | 67.4 | 70.3 | 67.8 | 58.0 | 49.4 | 61.2 |
| + CoLA (Ours) | 55.2 | 60.0 | 70.9 | 69.3 | 59.5 | 56.3 | 68.8 | 70.9 | 70.2 | 68.1 | 59.7 | 55.3 | **63.7** |
| DeYO [25] | 56.3 | 49.9 | 68.1 | 67.8 | 55.6 | 46.7 | 67.2 | 69.0 | 71.1 | 68.8 | 51.1 | 4.3 | 56.3 |
| + CoLA (Ours) | 56.2 | 55.1 | 71.2 | 70.2 | 54.8 | 54.5 | 70.0 | 71.5 | 71.0 | 69.0 | 53.7 | 54.3 | 62.6 |

## 4.1 Comparison with state-of-the-art methods

**Results under lifelong test-time adaptation.** We evaluate the long-term effectiveness of our CoLA on ImageNet-C [44] within a challenging lifelong TTA scenario where the model is online adapted to 15 corruptions over 10 rounds (total 150 corruptions), and the parameters will never be reset. We put more details of the experimental protocol in Appendix C due to page limits. From Table 1, we derive the following observations. 1) Our CoLA achieves new state-of-the-art results on the first round, last round, and the average of adaptation, suggesting our superiority. 2) Most methods, including CoTTA [56] and EATA [38] with an anti-forgetting strategy, experience performance degradation as the number of adaptation rounds increases (*e.g.*, ETA's performance degrades from 61.4% to 35.1% on the average accuracy), indicating the difficulty of the evaluated scenario. 3) By integrating our CoLA with existing methods, we enhance the performance steadily with more adaptations, demonstrating our effectiveness in accumulating and exploiting learned knowledge for long-term adaptation. 4) Although EATA mitigates performance degradation by introducing an anti-forgetting regularization, it suffers from the *stability-plasticity* trade-off, *i.e.*, the average accuracy drops from 61.4% (ETA) to 60.4% (EATA) in the first round. In contrast, our CoLA enhances ETA's performance even at the first round of adaptation, *i.e.*, 61.4% (ETA) *vs.* 62.0% (ETA+CoLA), indicating that CoLA does not limit the learning ability. The sensitivity analyses on threshold $z$ are provided in Appendix E.

**Results under collaborative test-time adaptation.** To evaluate our CoLA under the collaborative TTA scenario, we first assess its performance across multiple principal resource-abundant devices. From Table 2, our CoLA outperforms the integrated baseline from the adaptation to the second group of corruptions, *e.g.*, the accuracy of 58.0% (SAR+CoLA) *vs.* 54.4% (SAR) on 'Blur' in Devices 1. Moreover, this improvement becomes increasingly more pronounced as more knowledge is shared across devices, *e.g.*, improving the accuracy from 47.7% (SAR) to 55.5% (SAR+CoLA) on 'Noise' in Device 3. This demonstrates our effectiveness in facilitating knowledge sharing and exploitation across principal devices via our knowledge reprogramming learning scheme, *i.e.*, with Eqn. (2).

Table 3: Comparison on ImageNet-C (level 5) regarding **Accuracy (%)** under lifelong adaptation on **resource-limited follower devices**. T3A* resets the model after adaptation on each corruption. CoLA exploits the learned weights from Table 2 (*e.g.*, ETA + CoLA at the 7-th row) for Eqn. (3).

| Method | Noise | | | Blur | | | | Weather | | | | Digital | | | | Avg. |
|---|---|---|---|---|---|---|---|---|---|---|---|---|---|---|---|---|
| | Gaus. | Shot | Imp. | Def. | Glass | Mot. | Zoom | Snow | Frost | Fog | Brit. | Contr. | Elas. | Pix. | JPEG | |
| NoAdapt | 9.5 | 6.7 | 8.2 | 29.0 | 23.4 | 33.9 | 27.1 | 15.9 | 26.5 | 47.2 | 54.7 | 44.1 | 30.5 | 44.5 | 47.8 | 29.9 |
| T3A [21] | 9.5 | 7.0 | 8.7 | 23.3 | 23.3 | 31.2 | 25.9 | 11.9 | 24.2 | 44.0 | 52.2 | 41.0 | 30.1 | 43.0 | 47.0 | 28.2 |
| T3A* [21] | 9.5 | 6.5 | 8.1 | 29.8 | 24.1 | 34.3 | 28.2 | 16.0 | 26.9 | 49.0 | 55.5 | 44.5 | 33.1 | 44.5 | 48.2 | 30.5 |
| LAME [3] | 9.3 | 6.5 | 8.0 | 28.6 | 23.0 | 33.3 | 26.6 | 15.2 | 26.0 | 45.9 | 54.1 | 43.6 | 29.3 | 44.0 | 47.4 | 29.4 |
| CoLA (SAR) | 55.2 | 56.0 | 56.8 | 57.3 | 49.1 | 59.9 | 58.5 | 65.8 | 65.8 | 72.2 | 77.1 | 66.2 | 65.9 | 72.2 | 69.4 | 63.2 |
| CoLA (ETA) | 55.7 | 57.3 | 56.9 | 58.5 | 46.2 | 59.4 | 63.4 | 69.1 | 66.5 | 73.1 | 77.6 | 66.3 | 69.2 | 73.1 | 69.9 | **64.1** |
| CoLA (DeYO) | 56.6 | 57.7 | 57.5 | 58.2 | 47.7 | 55.5 | 39.0 | 69.6 | 67.2 | 73.5 | 78.0 | 67.0 | 70.4 | 73.5 | 70.3 | 62.8 |

Table 4: Comparison under single-domain TTA (on one principal device) w.r.t. **Acc (%)**. Results are averaged over 15 corruptions on ImageNet-C (level 5). **L.S** denotes label distribution shifts, **M.S** denotes mixed domain shifts per SAR [39].

| Method | Mild | L.S. | M.S | Avg. |
|---|---|---|---|---|
| NoAdapt | 29.9 | 29.9 | 29.9 | 29.9 |
| SAR [39] | 54.5 | 56.7 | 57.1 | 56.1 |
| + CoLA (Ours) | 57.7 | 58.5 | 58.0 | 58.1 |
| ETA [38] | 63.3 | 47.6 | 57.4 | 56.1 |
| + CoLA (Ours) | 64.4 | 55.2 | 58.3 | 59.3 |
| DeYO [25] | 64.1 | 61.3 | 59.1 | 61.5 |
| + CoLA (Ours) | 64.7 | 63.5 | 59.3 | 62.5 |

Table 5: Comparison w.r.t. wall-clock time and memory on ImageNet-C (Gaussian, level 5) on an A100 GPU. C/R refers to accuracy on ImageNet-C/R. **BP** is short for back-propagation. CoLA utilizes weights of ETA+CoLA in Table 2.

| Method | BP | C | R | T. (s) | Mem. (MB) |
|---|---|---|---|---|---|
| NoAdapt | ✗ | 9.5 | 43.1 | 50 | 816.6 |
| T3A [21] | ✗ | 9.5 | 42.1 | 158 | 909.9 |
| CoLA (Eqn. 3) | ✗ | 55.7 | 51.5 | 51 | 821.9 |
| EATA [38] | ✔ | 49.5 | 56.8 | 113 | 7439.3 |
| SAR [39] | ✔ | 44.0 | 51.8 | 202 | 7429.9 |
| ETA [38] | ✔ | 51.9 | 57.5 | 109 | 7429.6 |
| + CoLA (Eqn. 2) | ✔ | 54.3 | 59.0 | 112 | 7435.3 |

Given learned knowledge from resource-abundant principal devices (totaling 34 weights occupying 5.0 MB), we further evaluate the effectiveness of CoLA on resource-limited follower devices. From Table 3, we observe that existing TTA methods struggle to improve the performance of the source model without model updates, highlighting the urgent need for a more effective solution. In contrast, by exploiting shared knowledge adaptively in a forward-only manner with Eqn. (3), CoLA achieves a substantial performance gain, *e.g.*, enhancing the average accuracy from 29.9% to 64.1% in CoLA (ETA). Note that we also verify CoLA's sample efficiency as well as its computation and memory efficiency in Figure 3 (a) and Table 5. These results collectively underscore the importance of cross-device collaboration and our effectiveness regarding the scenario of collaborative TTA.

**Results under single-domain test-time adaptation.** Following DeYo [25], we validate our CoLA in both the wild scenario (*i.e.*, imbalanced label distribution shifts and mixture of distribution shifts) and the mild scenario of single-domain TTA, where the model is reset post-adaptation to each corruption. Here, CoLA saves learned weights for every adaptation to 10 batches of samples while maintaining a maximum of 32 weights (totaling 4.7 MB) by discarding the unused ones according to $\alpha_i$.

From Table 4, within all evaluated scenarios, incorporating CoLA consistently improves the performance by a considerable margin (*e.g.*, +2.0% on SAR w.r.t. overall average accuracy). Interestingly, the enhancement from CoLA may even help surpass a stronger baseline, *e.g.*, the average accuracy of 64.4% (ETA+CoLA) *vs.* 64.1% (DeYO) on the mild scenario, demonstrating our effectiveness. This improvement mainly stems from our ability to alleviate error accumulation. Given multiple saved weights, instead of naively selecting the newest weight that may have adapted to noise, CoLA dynamically favors the more optimal one via loss optimization. This renders CoLA more robust to scenarios where perturbations may occur. We also visualize $\alpha_i$ in Appendix E to offer more insights.

## 4.2 Ablation studies and more discussions

**Effectiveness of $T_l$ on sample efficiency in Eqn. (2).** Sample efficiency is particularly important in scenarios where the availability of target data is limited or early adaptation performance is paramount. As shown in Figure 3 (a), by leveraging the learned knowledge from other devices, ETA+CoLA achieves an up to $78.0\times$ speed up compared with ETA, *i.e.*, 51.7% accuracy with 640 samples (ETA+CoLA) *vs.* 51.37% accuracy with 49,920 samples (ETA), demonstrating the importance

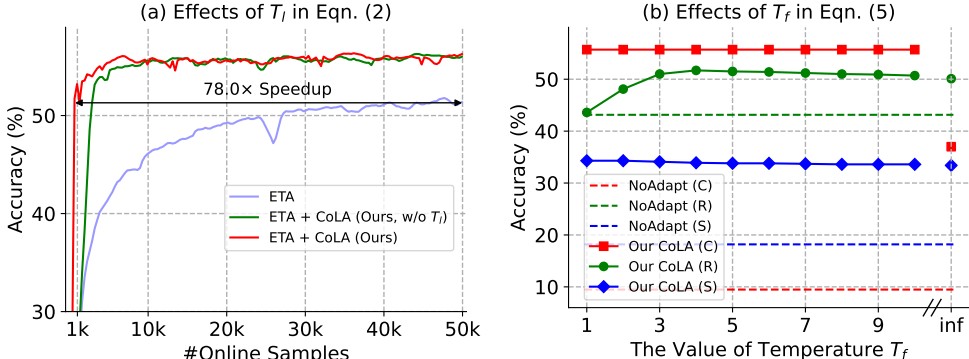

Figure 3: Ablation study of CoLA. In left, we compare the sample efficiency on Device2 in Table 2, Gaussian. Model's accuracy is recorded on the entire test set after adapting to $N$ test samples. In right, we evaluate the effectiveness of $T_f$ on seen (*i.e.*, ImageNet-C, Gaussian) and unseen distributions (*i.e.*, ImageNet-R/Sketch), where CoLA exploits weights of ETA+CoLA in Table 2 for Eqn. (3).

Table 6: Effectiveness of CoLA (Eqn. 2) on **unseen distributions** with weights learned on ImageNet-C from Table 2.

| Method | R | S | Avg. |
|---|---|---|---|
| SAR [39] | 51.9 | 33.8 | 42.8 |
| + CoLA (Ours) | 55.1 | 39.3 | 47.2 |
| ETA [38] | 57.7 | 43.1 | 50.4 |
| + CoLA (Ours) | 59.0 | 43.2 | 51.1 |

Table 7: Effectiveness of CoLA on **prompt tuning**. Results are reported on ImageNet and its variants with CLIP-RN50 [43]. CoLA exploits 78 hard prompts for Eqn. (2).

| Method | I | A | V2 | R | S | Avg. |
|---|---|---|---|---|---|---|
| NoAdapt | 58.2 | 21.8 | 51.4 | 56.2 | 33.4 | 44.2 |
| Ensemble | 59.8 | 23.2 | 52.9 | 60.7 | 35.5 | 46.4 |
| TPT [47] | 60.7 | 26.7 | 54.7 | 59.1 | 35.1 | 47.3 |
| + CoLA (Ours) | 62.2 | 28.0 | 55.4 | 60.4 | 34.7 | 48.1 |

of cross-device collaboration and our effectiveness to facilitate knowledge sharing and utilization. Moreover, upon integrating $T_l$ in Eqn. (2), CoLA demonstrates superiority without necessitating too many test samples, *i.e.*, 53.2% accuracy (ETA+CoLA) *vs.* 22.2% accuracy (ETA+CoLA w.o. $T_l$) with 960 samples, suggesting the effectiveness of $T_l$ to promote swift adaptation under distribution shifts. We also provide more comparison in Appendix E regarding sample efficiency on unseen distributions.

**Effectiveness of $T_f$ on robustness in Eqn. (5).** We evaluate the robustness of Eqn. (3) on seen/unseen distributions (*i.e.*, whether resource-abundant devices have encountered the evaluated distribution). From Figure 3 (b), our CoLA consistently outperforms NoAdapt regardless of $T_f$, demonstrating our effectiveness. On seen distribution, our improvement is particularly significant while the performance is insensitive to $T_f$ in a reasonable range. However, when $T_f$ is set to infinity (*i.e.*, averaging different weights), performance experiences degradation as appropriate knowledge plays a less significant role. On unseen distribution (*e.g.*, ImageNet-R), aggregating a minority of knowledge may be insufficient to address distribution shifts. In this case, $T_f$ plays an important role in enhancing robustness by aggregating the strengths of diverse knowledge. We fix $T_f$ to 5 in experiments without careful tuning.

**Memory and computation consumption of CoLA.** Besides achieving strong performance across various scenarios, we demonstrate that CoLA incurs negligible computation and memory costs. From Table 5, on resource-limited follower devices, CoLA (Eqn. 3) substantially outperforms T3A in terms of accuracy and runtime memory (*i.e.,* 55.7% *vs.* 9.5%, and 821.9 MB *vs.* 909.9 MB) while maintaining nearly the same efficiency as NoAdapt. On resource-abundant principal devices, CoLA enhances the performance of ETA by a considerable margin (*i.e.*, +2.4% on ImageNet-C) while incurring only 5.7 MB of additional runtime memory and 3s of latency, indicating that CoLA is even lighter than the regularizer introduced in EATA. Note that CoLA (Eqn. 3) determines appropriate re-weighting for different knowledge with a batch of test samples, outperforming ETA+CoLA w.r.t average accuracy on ImageNet-C. This underscores the potential of collaborative TTA in real world, where a device may adapt effectively using only negligible costs given adequate shared knowledge. We also show that CoLA can efficiently scale to over 10,000 domain vectors in Appendix E.

**Generalization of Eqn. (2) on unseen distributions.** Following Figure 3 (b), we further validate the effectiveness of CoLA (Eqn. 2) on unseen distributions. From Table 6, CoLA consistently enhances performance on unseen distributions, yielding a notable 4.4% improvement on SAR in terms of

average accuracy. Such improvement can be attributed to the transferability of knowledge learned from similar domains [63]. These findings collectively indicate that the effectiveness of CoLA is not limited to previously encountered distributions on both principal agents and follower agents.

**Prompt tuning with CoLA using multiple hard prompts.** Besides aggregating learned knowledge from other devices, we demonstrate that CoLA can also benefit from aggregating diverse knowledge from humans (*i.e.*, manual-designed hard prompts). As shown in Table 7, compared with TPT which is limited to leveraging a single hard prompt, CoLA enhances the adaptation performance on 4 out of 5 datasets (*e.g.*, +1.5% on ImageNet w.r.t. accuracy). These results collectively indicate that CoLA effectively exploits both the knowledge of humans and the knowledge from optimization, which may bring new perspectives to the design of learning algorithms when introducing diverse human prior knowledge is beneficial, *e.g.*, chain-of-thoughts [58]. All used prompts are listed in Appendix C.

**Differences and advantages over federated TTA [1, 23].** The main difference is that our CoLA conducts collaborative learning at the testing phase, whereas federated TTA conducts collaborative learning during federated source training. For instance, FedTHE+ [23] federatedly trains a global and a local personalized model for each client, then adaptively ensemble their outputs at test time. ATP [1] federatedly learns module-specific adaptation rates across clients during training for test-time adaptation. However, these methods still conduct TTA independently on each devices during testing, and thus inherits the limitation of the single-device TTA methods. Moreover, in federated TTA [1, 23], the training phase and test-time adaptation phase are highly correlated, which means they can only use their own federated-trained models during TTA. This makes these methods restricted for real-world applications. In contrast, our CoLA enhances test-time model adaptation performance and efficiency by leveraging knowledge from multiple devices in the application environment, which essentially establishes a new unsupervised on-time TTA paradigm. Moreover, our CoLA paradigm can be applied to any pre-trained models, and thus offers much better flexibility in deployment. Additional comparisons with FedAvg [32] for collaborative adaptation are also provided in Appendix E.

# 5 Conclusion

In this paper, we propose a multi-device **Co**llaborative **L**ifelong **A**daptation (CoLA) paradigm for test-time adaptation (TTA), which addresses a practical scenario where multiple devices with different computational resources and latency requirements need to perform TTA simultaneously. In particular, we first accumulate a set of shared domain knowledge vectors with an efficient domain shift detector. Based on this, we develop a knowledge reprogramming learning strategy on principal agents, which leverages backpropagation-based optimization to aggregate existing knowledge while learning new domain-specific parameters simultaneously. To further improve adaptation efficiency, we introduce an optimization-free TTA strategy on follower agents, which solely aggregates the shared domain vectors based on domain similarity. In CoLA, all devices/agents work collaboratively while keeping privacy preserved and communication efficient. Experiments verify that CoLA boosts the performance and efficiency of existing TTA solutions in collaborative, lifelong, and single-domain TTA scenarios.

# Acknowledgments

This work was partially supported by National Natural Science Foundation of China (NSFC) 62072190 and TCL Science and Technology Innovation Fund. The authors thank Jinwu Hu and Yu Hu for discussions on domain knowledge reprogramming, and Yaofo Chen for consultations on cloud-edge test-time adaptation.

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

# Supplementary Materials for
# " Cross-Device Collaborative Test-time Adaptation "

## Contents

Table A: Characteristics of problem settings that adapt a trained model to a potentially shifted domain. 'Online' adaptation predicts a single or batch of incoming test samples immediately. '#Devices' is the number of devices involved in TTA. 'Learned Knowledge' is the knowledge from model adaptation.

| Setting | Target Data | Testing Loss | Online | #Devices | Learned Knowledge | Resource-Limited Devices | Privacy | Data Transmission |
|---|---|---|---|---|---|---|---|---|
| Fine-tuning | $\mathbf{x}^t, y^t$ | – | ✗ | 1 | Not Considered | Not Considered | – | – |
| Continual learning | $\mathbf{x}^t, y^t$ | – | ✗ | 1 | Accumulated | Not Considered | – | – |
| Unsupervised DA | $\mathbf{x}^t$ | – | ✗ | 1 | Accumulated | Not Considered | – | – |
| Test-time training | $\mathbf{x}^t$ | $\mathcal{L}(\mathbf{x}^t)$ | ✓ | 1 | Not Considered | Not Considered | – | – |
| Fully TTA | $\mathbf{x}^t$ | $\mathcal{L}(\mathbf{x}^t)$ | ✓ | 1 | Not Considered | Partly Applicable | – | – |
| Federated TTA [1, 23] | $\mathbf{x}^t$ | $\mathcal{L}(\mathbf{x}^t)$ | ✓ | 1 | Not Considered | Not Considered | Preserved | Intensive |
| Cloud-Edge TTA [4] | $\mathbf{x}^t$ | $\mathcal{L}(\mathbf{x}^t)$ | ✓ | 2 | Not Considered | Applicable | Violated | Intensive |
| CoLA (Ours) | $\mathbf{x}^t$ | $\mathcal{L}(\mathbf{x}^t)$ | ✓ | $M$ | Accumulated | Applicable | Preserved | Intermittent |

# A    Related work

We summarize our main differences in Table A and discuss the related works in the following.

**Test-time adaptation** (TTA) seeks to enhance the model performance on unseen, potentially shifted test data, by directly learning from the test data itself. We categorize the related TTA works into the following four groups for discussion, according to 1) the number of devices involved in adaptation; 2) their dependence on backward propagation; and 3) the availability of multiple models.

• *Single-device backpropagation-based TTA.* Test-Time Training (TTT) [51] first proposes this pipeline. During the training phase, TTT methods train a source model with both a supervised and a self-supervised branch. Given a test sample during testing, they typically update the shared encoder with the self-supervised objectives, such as rotation prediction [51], contrastive learning [30, 2], reconstruction learning [11]. To avoid altering the model training phase and access to source data, Fully TTA methods directly update an on-the-fly model via unsupervised learning objectives, including, but not limited to, entropy minimization [54, 39], prediction consistency maximization [64, 10] and feature distribution alignment [34].

In pursuit of efficient backpropagation-based TTA, the attempts of existing methods can be generally categorized into: 1) *Sample Efficiency*. As test data are not equally important for adaptation, some recent works [38, 39, 47, 25] have devised various sample selection strategies to identify reliable and non-redundant samples for test-time learning. It reduces the noise in the gradient and the number of samples for TTA, thereby enhancing adaptation performance and efficiency. 2) *Memory Efficiency*: EcoTTA [48] reduces run-time memory by optimizing only the parameter-efficient adapters. MECTA [19] reduces the batch size at testing, while it further proposes a domain-aware batch normalization layer to stabilize TTA using only a small batch size.

Nevertheless, these methods focus on single-device adaptation, where all devices adapt from scratch. In this sense, valuable knowledge learned from other devices is neglected, damaging the adaptation performance and efficiency. Moreover, these methods still rely on computationally intensive backpropagation for model updates, which hinders their applicability in resource-limited devices or latency-sensitive scenarios. To address this, we propose a gradient-based and forward-only collaboration paradigm, facilitating knowledge accumulation, sharing, and utilization across devices.

Recently, TTA-CDKM [49] proposes to learn and reuse multiple groups of model parameters for better adaptation, with each group representing the knowledge of a domain. However, TTA-CDKM exploits only a parameter group during inference, and thus fails to aggregate the strength of diverse domain knowledge. On the other hand, TTA-CDKM updates the stored parameters when adapting to each batch of test samples, which introduces intensive communication costs for weight synchronization that are impractical for multi-device collaboration. In contrast, our CoLA effectively learns to aggregate diverse strengths of domain vectors at testing while ensuring communication efficiency.

• *Single-device forward-only TTA.* In the development of BP-free TTA, early research mostly focused on calibrating the statistics of batch normalization layers by leveraging the test data to estimate test statistics [36, 46]. Nevertheless, these strategies only conduct adaptation on batch normalization layers, limiting their applicability to various architectures. In pursuit of a more general forward-only TTA solution, existing methods can be generally divided into: 1) *Input-Level Adaptation*,

where the corrupted test images are reconstructed before making predictions based on a diffusion model [12, 40]. 2) *Output-Level Adaptation*, where T3A [21] proposed a prototype-based classifier for adaptive predictions, and LAME [3] directly corrects the predicted logits. However, since BP-free TTA does not leverage model feedback, *i.e.*, un/self-supervised learning objectives, for knowledge acquisition, it often results in suboptimal performance when dealing with out-of-distribution testing data. In our paper, we address this by facilitating knowledge sharing and utilization across devices, where we adaptively aggregate the learned knowledge of other devices in a forward-only manner.

• *Single-device test-time aggregation.* Given multiple pretrained models, one can leverage unsupervised objectives at test time to adaptively aggregate their diverse strengths. Based on the level on which aggregation is performed, existing solutions can be generally divided into:

1) *Output-wise test-time aggregation*: Early research mainly focuses on output-wise test-time aggregation, where the aggregation is conducted on the output logit of each model. EMEA [57] first proposed this pipeline. Given multiple models trained on different datasets or different label distributions, this paradigm introduced a reweighting vector to aggregate the outputs logit of various models during testing, where the reweighting vector is optimized based on entropy minimization [57, 65], consistency loss [5], *etc.* More Recently, Mute [9] jointly updates the reweighting vector and all pretrained models at test time. Nevertheless, these strategies necessitate forward passes for each candidate model, *i.e.*, $\mathcal{O}(N)$ forward passes with $N$ the number of candidate models. Thus, they typically consume substantially more computation power, impeding their feasibility at edge devices.

2) *Parameter-wise test-time aggregation*. In contrast, parameter-wise test-time aggregation methods directly merge several candidate parameters into a single model without necessitating additional forward passes, *i.e.*, $\mathcal{O}(1)$ forward passes neglect of $N$, rendering it more efficient in computation. To this end, Adamerging [62] first introduces the learnable parameter-wise weighting vectors, which aggregate various models before performing forward passes. Building upon this, WEMOE [53] further introduces the mixture of experts (MoE) into this paradigm, where the parameters from different models are considered as different experts, while the parameter-wise weighting vector is generated by a router that is trained during testing, with the entropy minimization objective.

Nevertheless, existing test-time aggregation solutions assume the availability of pretrained models from the target domain, which struggles to fulfill the conventional TTA settings. Unlike these methods, we perform both new knowledge learning and existing knowledge aggregation simultaneously at test time, where knowledge is accumulated across all devices from previous learning.

• *Multi-device test-time adaptation.* Existing TTA methods typically conduct adaptation on a single device, where multi-device test-time adaptation is heavily overlooked. In recent academics, FedTHE+ [22] proposes a global-local scheme for federated test-time adaptation. Specifically, during each round of federated training, each device trains the global model on the local data, where the trained model is then sent back to the server. During testing, it adaptively merges the output logit of the global and local model based on entropy minimization and feature alignment loss. Nevertheless, it necessitates alteration to model training, limiting their applicability to scenarios where training data is unavailable. ATP [1] obtains module-specific adaptation rates via federated learning across clients and applies the adaptation rates in TTA. However, these methods still conduct TTA independently on each devices, and thus inherits the limitation of the single-device TTA method. More recently, CEMA [4] proposes a cloud-edge collaboration paradigm for TTA. In CEMA, edge devices perform pure inference to filter reliable and informational samples to reduce communication burden, where the computationally intensive model updates are offloaded to the cloud server. Nevertheless, CEMA necessitates intensive transmission of both data and model weights, which introduces a heavy communication burden and may violate user privacy. Unlike these methods, we conducts collaborative adaptation at test time and necessitate only the intermittent transmission of updated model parameters, which is more practical in real-world implementation.

**Unsupervised domain adaptation (UDA).** Conventional UDA tackles distribution shifts by jointly optimizing a source model on both labeled source data and unlabeled target data to learn domain-invariant features [31, 41, 45, 67, 66]. To avoid access to source data, recently CPGA [42] generates feature prototypes for each category with pseudo-labeling. SHOT [27] learns a target-specific feature extractor by information maximization for representation alignment. Nevertheless, these methods necessitate pre-collected target datasets for offline adaptation, which limits their applicability in the real world. In contrast, our method adapts in an online manner and does not access the model training phase or access to source data, which facilitates a more practical adaptation paradigm.

# B   More design details of CoLA

In the section, we further elaborate more details of our methods regarding the design of saving domain knowledge upon distribution shifts

**Initialization of $\alpha$ and $\Delta\theta$ in Eqn. (2) for stable adaptation warm-up.** Since ground-truth labels are absent in our online TTA scenario to provide stable learning signals for Eqn. (2), a careful initialization of $\alpha$ and $\Delta\theta$ is crucial upon distribution changes. In this sense, to ensure the stability of model performance pre and post encountering a new domain while enabling flexible aggregation of existing knowledge, we devise the initialization strategy by considering the following requirements: 1) the overall model weights $\theta_l^{n+1}$ initialized after encountering the domain $n+1$ should remain the same as $\theta_l^n$ that learned on the $n$-th domain; 2) $\Delta\theta$ should be as small as possible to enable flexible adaptation, *e.g.*, when the new samples are all from in-distribution, one can obtain the solution of $\theta_l = \theta_l^o$ by setting $\alpha_0 = 1$ if $\Delta\theta$ is negligible; 3) for $\alpha = \mathrm{softmax}(\beta)$, the initial $\beta$ should not be very sharp, *e.g.*, $\beta = [0, \inf, 0, ..., 0]$, as it will hinder the model learning from selecting diverse knowledge.

Formally, let $\{\Delta\theta^{(i)}\}_{i=1}^L$ be the learned parameter on the $i$-th layer in $\Delta\theta$. We define $\Delta\theta$'s magnitude as the maximum scale of parameters across different layers $\xi = \max\{\frac{1}{J}\sum_{j=1}^J |\Delta\theta^{(i,j)}|\}_{i=1}^L$, where $J$ is the number of parameters in each layer. To this end, we satisfy the above constraint by defining the reweighting logit $\Delta\theta_{n+1}$ as:

$$\beta_{n+1} = \frac{1}{T_l} \ln\left((s-1)\sum_{i=0}^n e^{\beta_i T_l}\right), \quad s = \max\{1, \frac{\xi}{w_m}\} \tag{8}$$

Here, $\beta_{n+1}$ is designed so that the magnitude $\xi'$ of the recalculated $\Delta\theta$ satisfies $\xi' \leq w_m$, where $w_m$ is a constrained value set to 0.01. Moreover, to improve numerical stability in case $T_l$ is 0, we clip the value of $\beta_{n+1}$ between $[-10, 10]$. The preserved knowledge $\Delta\theta_{n+1}$ is then shared across devices.

We next provide a proposition to validate the reasonableness of the design for $\beta_{n+1}$:

**Proposition 1.** *Given $\beta_{n+1}$ in Eqn. (8), the constraints that $\theta_l^{n+2} = \theta_l^{n+1}$ and $\xi' \leq w_m$ holds.*

*Proof.* Assume that the model has learned from $n$ previously domains. According to Eqn. (2), when the model adapts on the $(n+1)$-th domain, we have

$$\theta_l^{n+1} = \theta_l^o + \alpha_0\Delta\theta_0 + \alpha_1\Delta\theta_1 + \cdots + \alpha_n\Delta\theta_n + \Delta\theta, \quad \text{where } \alpha = \mathrm{softmax}(\beta \cdot T_l).$$

When encountering the $(n+2)$-th domain, we should save learned knowledge $\Delta\theta$ from the $(n+1)$-th domain. To adaptively aggregates prior knowledge and saved knowledge, we initialize $\alpha_{n+1}$ for $\Delta\theta_{n+1}$. Besides, we initialize $\Delta\theta'$ in order to learn new knowledge from the $(n+2)$-th domain. So we have

$$\theta_l^{n+2} = \theta_l^o + \alpha_0'\Delta\theta_0 + \alpha_1'\Delta\theta_1 + \cdots + \alpha_n'\Delta\theta_n + \alpha_{n+1}\Delta\theta_{n+1} + \Delta\theta'$$

After adding $\alpha_{n+1}$, to maintain the vector $\alpha$ as normalized weights, $\alpha_i$ will change to

$$\begin{aligned}
\alpha_i' &= \frac{e^{\beta_i T_l}}{(\sum_{j=0}^n e^{\beta_j T_l}) + e^{\beta_{n+1} T_l}} \\
&= \frac{e^{\beta_i T_l}}{\sum_{j=0}^n e^{\beta_j T_l}} \cdot \frac{\sum_{j=0}^n e^{\beta_j T_l}}{(\sum_{j=0}^n e^{\beta_j T_l}) + e^{\beta_{n+1} T_l}} \\
&= \alpha_i \cdot \frac{\sum_{j=0}^n e^{\beta_j T_l}}{(\sum_{j=0}^n e^{\beta_j T_l}) + e^{\beta_{n+1} T_l}}.
\end{aligned}$$

As we want to ensure the stability of model performance, the overall model weights $\theta_l^{n+2}$ should be the same as $\theta_l^{n+1}$ that learned on the $(n+1)$-th domain, so $\Delta\theta_{n+1} = \theta_l^{n+2} - \theta_l^o = \theta_l^{n+1} - \theta_l^o$.

Thus, we expand $\boldsymbol{\theta}_l^{n+2}$ as:

$$\boldsymbol{\theta}_l^{n+2} = \boldsymbol{\theta}_l^o + \alpha_0' \Delta\boldsymbol{\theta}_0 + \alpha_1' \Delta\boldsymbol{\theta}_1 + \cdots + \alpha_n' \Delta\boldsymbol{\theta}_n + \alpha_{n+1} \Delta\boldsymbol{\theta}_{n+1} + \Delta\boldsymbol{\theta}'$$

$$= \boldsymbol{\theta}_l^o + \frac{\sum_{i=0}^n e^{\beta_i T_l}}{\left(\sum_{i=0}^n e^{\beta_i T_l}\right) + e^{\beta_{n+1} T_l}} \left(\alpha_0 \Delta\boldsymbol{\theta}_0 + \alpha_1 \Delta\boldsymbol{\theta}_1 + \cdots + \alpha_n \Delta\boldsymbol{\theta}_n\right)$$

$$+ \frac{e^{\beta_{n+1} T_l}}{\left(\sum_{i=0}^n e^{\beta_i T_l}\right) + e^{\beta_{n+1} T_l}} \cdot \Delta\boldsymbol{\theta}_{n+1} + \Delta\boldsymbol{\theta}'$$

$$= \boldsymbol{\theta}_l^o + \frac{\sum_{i=0}^n e^{\beta_i T_l}}{\left(\sum_{i=0}^n e^{\beta_i T_l}\right) + e^{\beta_{n+1} T_l}} \left(\boldsymbol{\theta}_l^{n+1} - \boldsymbol{\theta}_l^o - \Delta\boldsymbol{\theta}\right)$$

$$+ \frac{e^{\beta_{n+1} T_l}}{\left(\sum_{i=0}^n e^{\beta_i T_l}\right) + e^{\beta_{n+1} T_l}} \cdot \left(\boldsymbol{\theta}_l^{n+1} - \boldsymbol{\theta}_l^o\right) + \Delta\boldsymbol{\theta}'$$

$$= \boldsymbol{\theta}_l^{n+1} - \frac{\sum_{i=0}^n e^{\beta_i T_l}}{\left(\sum_{i=0}^n e^{\beta_i T_l}\right) + e^{\beta_{n+1} T_l}} \cdot \Delta\boldsymbol{\theta} + \Delta\boldsymbol{\theta}'.$$

Considering $\boldsymbol{\theta}_l^{n+2} = \boldsymbol{\theta}_l^{n+1}$, we have

$$\boldsymbol{\theta}_l^{n+1} - \frac{\sum_{i=0}^n e^{\beta_i T_l}}{\left(\sum_{i=0}^n e^{\beta_i T_l}\right) + e^{\beta_{n+1} T_l}} \cdot \Delta\boldsymbol{\theta} + \Delta\boldsymbol{\theta}' = \boldsymbol{\theta}_l^{n+1}$$

$$\Rightarrow \quad \Delta\boldsymbol{\theta}' = \frac{\sum_{i=0}^n e^{\beta_i T_l}}{\left(\sum_{i=0}^n e^{\beta_i T_l}\right) + e^{\beta_{n+1} T_l}} \cdot \Delta\boldsymbol{\theta}.$$

As $\Delta\boldsymbol{\theta}'$ should be as small as possible to enable flexible adaptation, the magnitude of $\Delta\boldsymbol{\theta}'$ should be no larger than $w_m$, i.e. $\xi' \leq w_m$. So we scale $\Delta\boldsymbol{\theta}$ by a factor of $s$, i.e. $\Delta\boldsymbol{\theta}' = \frac{1}{s} \cdot \Delta\boldsymbol{\theta}$. If the magnitude of $\xi'$ already satisfies $\xi' \leq w_m$, we set $s = 1$. Otherwise, we set $s = \frac{\xi}{w_m}$.

Thus, we have $s = \max\{1, \frac{\xi}{w_m}\}$ and

$$\frac{\sum_{i=0}^n e^{\beta_i T_l}}{\left(\sum_{i=0}^n e^{\beta_i T_l}\right) + e^{\beta_{n+1} T_l}} = \frac{1}{s}$$

$$\Rightarrow \quad \beta_{n+1} = \frac{1}{T_l} \ln\left((s-1) \sum_{i=0}^n e^{\beta_i T_l}\right).$$

$\square$

# C More implementation details

## C.1 More details on datasets

In this paper, we conduct experiments on ImageNet-1K [6] and its five variants to evaluate the out-of-distribution generalization ability, *i.e.*, ImageNet-C [16], ImageNet-R [15], ImageNet-A [17], ImageNet-V2 [44], and ImageNet-Sketch [55].

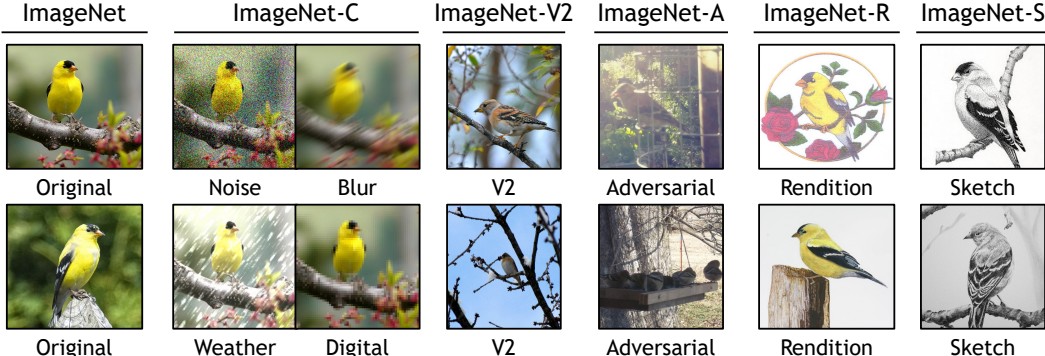

Figure A: Visualizations of images in ImageNet and ImageNet-C/V2/A/R/Sketch.

**ImageNet-C** consists of various versions of corruption applied to 50,000 validation images from ImageNet. The dataset encompasses 15 distinct corruption types of 4 main groups, including Gaussian noise, shot noise, impulse noise, defocus blur, glass blur, motion blur, zoom blur, snow, frost, fog, brightness, contrast, elastic transformation, pixelation, and JPEG compression. Each corruption type is characterized by 5 different levels of severity, with higher severity levels indicating a more severe distribution shift. In our experiments, we specifically utilize severity level 5 for evaluation.

**ImageNet-R** contains 30,000 images featuring diverse artistic renditions of 200 ImageNet classes. These images are predominantly sourced from Flickr and filtered by Amazon MTurk annotators.

**ImageNet-A** comprises 7,500 images covering 200 ImageNet classes. These images are derived from real-world, naturally occurring examples that lead to a notable degradation in classifier performance.

**ImageNet-V2** is a newly collected test dataset extracted from the same test distribution as ImageNet. It comprises three test sets, each containing 10,000 new images and covering 1000 ImageNet classes. Following previous TTA methods [36], we utilize the Matched Frequency subset of ImageNet-V2 for evaluation, in which the images are sampled to match the class frequency distributions of the original ImageNet validation dataset.

**ImageNet-Sketch** consists of 50,899 images represented as black and white sketches, encompassing 1000 ImageNet classes. Each class contains approximately 50 images.

## C.2 More experimental protocols on evaluation

We use ViT-Base [8] as the source model for all experiments except Table 7. The model is trained on the source ImageNet-1K training set and we directly obtain the model weights[3] from `timm`[4] repository [60]. In Table 7, we adopts CLIP-RN50 [43] as the source model by following TPT [47]. The model weights[5] are directly obtained from the original CLIP[6] repository [43]. All experiments are conducted on a single NVIDIA A100 GPUS, using PyTorch framework with version 1.8.0.

**Evaluation on lifelong TTA.** In Table 1, the model is online adapted to 15 corruptions over 10 rounds (total 150 corruptions), where the parameters will never be reset. Specifically, the corruptions

---

[3]https://storage.googleapis.com/vit_models/augreg/B_16-i21k-300ep-lr_0.001-aug_medium1-wd_0.1-do_0.0-sd_0.0–imagenet2012-steps_20k-lr_0.01-res_224.npz

[4]https://github.com/pprp/timm

[5]https://openaipublic.azureedge.net/clip/models/afeb0e10f9e5a86da6080e35cf09123aca3b358a0c3e3b6c78a7b63bc04b6762/RN50.pt

[6]https://github.com/openai/CLIP

comes in the following order for each round: Gaussian Noise $\to$ Defocus Blur $\to$ Snow $\to$ Contrast $\to$ Shot Noise $\to$ Glass Blur $\to$ Frost $\to$ Elastic Transform $\to$ Impulse Noise $\to$ Motion Blur $\to$ Fog $\to$ Pixelate $\to$ Brightness $\to$ Zoom Blur $\to$ JPEG Compression. Here, subsequent corruptions differ in type, which poses a more significant challenge for TTA methods to leverage previously learned knowledge for adaptation.

**Evaluation on collaborative TTA.** In Table 2, we evaluate our effectiveness in collaborating on multiple resource-abundant principal devices, where the learned knowledge on each device is shared with others post-adaptation to each type of corruption. Specifically, the corruption order in each type of corruption is as follows. 1) `Noise`: Gaussian Noise $\to$ Shot Noise $\to$ Impulse Noise. 2) `Blur`: Defocus Blur $\to$ Glass Blur $\to$ Motion Blur $\to$ Zoom Blur. 3) `Weather`: Snow $\to$ Frost $\to$ Fog $\to$ Brightness. 4) `Digital`: Contrast $\to$ Elastic Transformation $\to$ Pixelate $\to$ JPEG Compression.

Then, given learned knowledge from Table 2, we further verify our effectiveness on resource-limited follower devices in Table 3, *e.g.*, CoLA (SAR) utilizes the learned weights of SAR+CoLA from Table 2, and CoLA (ETA) utilizes the learned weights of ETA+CoLA from Table 2. Here, the resource-limited follower devices adapt in a lifelong manner per EATA [38], where the corruptions come in the following order: Gaussian Noise $\to$ Shot Noise $\to$ Impulse Noise $\to$ Defocus Blur $\to$ Glass Blur $\to$ Motion Blur $\to$ Zoom Blur $\to$ Snow $\to$ Frost $\to$ Fog $\to$ Brightness $\to$ Contrast $\to$ Elastic Transformation $\to$ Pixelate $\to$ JPEG Compression.

**Evaluation on single-domain TTA.** In Table 4, we validate our CoLA in both the wild scenario (i.e., imbalanced label distribution shifts and mixed domain shifts) and the mild scenario of single-domain TTA, where the model is reset post-adaptation to each corruption. Here, imbalanced label distribution shifts denotes scenarios where test data come in a class order, mixed domain shifts denotes scenarios where test data are drawn from multiple randomly mixed domains with different distribution shifts.

**Evaluation on prompt tuning.** In Table 7, CoLA aggregates 78 diverse hard prompts with the class token at the end for Eqn. (2), where different prompts are padded to the same length by inserting empty spaces at the beginning. Note that adaptation is conducted in an episodic manner by following TPT [47], where we reset $\Delta\theta = 0$ post-adaptation to each test sample in Eqn. (2), while $\alpha$ is continually optimized without reset due to its stability benefit from normalization. Results in Table 7 further demonstrate our effectiveness across various TTA scenarios. The utilized hard prompts for Eqn. (2) are originally developed by [43], as listed below:

---
**List of Hard Prompts:**

a drawing of the {class}, art of a {class}, itap of the {class}, a drawing of a {class}, a origami {class}, a photo of a nice {class}, a blurry photo of a {class}, a close-up photo of the {class}, a photo of a clean {class}, a photo of a weird {class}, a photo of a small {class}, a photo of the large {class}, a pixelated photo of the {class}, a embroidered {class}, a photo of the clean {class}, the origami {class}, the plushie {class}, a photo of a cool {class}, a sculpture of the {class}, a low resolution photo of the {class}, a bad photo of the {class}, a jpeg corrupted photo of a {class}, a rendition of the {class}, a photo of the cool {class}, a low resolution photo of a {class}, a cropped photo of the {class}, the plastic {class}, a sculpture of a {class}, a pixelated photo of a {class}, itap of a {class}, a doodle of a {class}, a sketch of a {class}, a plastic {class}, itap of my {class}, a close-up photo of a {class}, a bright photo of a {class}, art of the {class}, graffiti of the {class}, a tattoo of a {class}, a sketch of the {class}, a dark photo of a {class}, a tattoo of the {class}, a photo of the dirty {class}, a black and white photo of the {class}, a photo of a {class}, a painting of the {class}, a cropped photo of a {class}, a photo of a large {class}, a photo of the weird {class}, graffiti of a {class}, a painting of a {class}, a cartoon {class}, the cartoon {class}, a good photo of the {class}, a jpeg corrupted photo of the {class}, a bad photo of a {class}, a photo of the small {class}, a rendering of the {class}, a photo of a dirty {class}, a rendition of a {class}, a blurry photo of the {class}, the toy {class}, the embroidered {class}, a rendering of a {class}, a photo of a hard to see {class}, a dark photo of the {class}, a doodle of the {class}, a good photo of a {class}, a photo of the {class}, a photo of many {class}, a plushie {class}, a photo of the nice {class}, a bright photo of the {class}, a toy {class}, a photo of the hard to see {class}, a photo of one {class}, a photo of my {class}, a black and white photo of a {class}, a sketch of a {class}

---

## C.3 More experimental protocols on methods

**CoLA (Ours).** For principal agents, We directly leverage the learnable test-time objectives from the integrated TTA methods, as listed below. In Eqn. (2), $\Delta\theta$ is optimized by following the update rules and the hyper-parameters of the integrated baseline, $\alpha$ and $T_l$ is updated via the AdamW optimizer with a learning rate of 0.1, and we introduce a weight decay of 0.1 on $\alpha$. We set the shift detection threshold $z$ to 0.1 for all experiments. Moreover, we introduce a weight decay on $\Delta\theta$ for Table 1 and Table K, *i.e.*, 0.1/0.4/0.4 for SAR/ETA/DeYO in Table 1 and 0/0.4/10 for EATA/ETA/SAR in Table K, respectively. This weight decay, which can be viewed as a simple implementation of the regularizer in EATA [38], helps learn compact and non-redundant knowledge in the domain vectors, thereby benefiting knowledge accumulation. For follower agents, we consistently set $T_f$ in Eqn. (3) to 5 for all experiments. Moving average factor $\lambda$ is set to 0.2.

**SAR[7] [39].** We follow all hyper-parameters that are set in SAR unless it does not provide. Specifically, we use SGD as the update rule, with a momentum of 0.9, batch size of 64, and a learning rate of 0.001. The entropy threshold $E_0$ is set to $0.4 \times \ln C$, where $C$ is the number of task classes. The trainable parameters are the affine parameters of the layer normalization layers from blocks 1 to blocks 8.

**ETA & EATA[8] [38].** We follow all hyper-parameters that are set in ETA/EATA unless it does not provide. Specifically, we use SGD as the update rule, with a momentum of 0.9, batch size of 64, and a learning rate of 0.001. The entropy threshold $E_0$ is set to $0.4 \times \ln C$, where $C$ is the number of task classes. For EATA, *i.e.*, ETA with an anti-forgetting regularizer, we use 2,000 samples to estimate the importance of each parameter. The trainable parameters are all affine parameters of layer normalization layers.

**DeYO[9] [25].** We follow all hyper-parameters that are set in SAR unless it does not provide. Specifically, we use SGD as the update rule, with a momentum of 0.9, batch size of 64, and a learning rate of 0.001. The entropy threshold $E_0$ is set to $0.4 \times \ln C$ and the entropy factor $\tau_{\text{Ent}}$ is set to $0.5 \times \ln C$, where $C$ is the number of task classes. The Pseudo-Label Probability Difference (PLPD) threshold $\tau_{\text{PLPD}}$ is set to 0.3 in Table 4, and 0.2 for other experiments by following the original paper. Trainable parameters are the affine parameters of the layer normalization layers from blocks 1 to blocks 8.

**CoTTA[10] [56].** We follow all hyperparameters that are set in CoTTA unless it does not provide. Specifically, we use SGD as the update rule, with a momentum of 0.9, and a batch size of 64. The learning rate is chosen from {0.05, 0.01, 0.005, 0.001, 0.0005, 0.0001} and the augmentation threshold $p_{th}$ is chosen from {0.1, 0.2}. In all experiments, we consistently set the learning rate to 0.001 and $p_{th}$ to 0.1 given the optimal accuracy observed in Table 1. For images below the threshold, we conduct 32 augmentations including color jitter, random affine, Gaussian blur, random horizontal flip, and Gaussian noise. The restoration probability of is set to 0.01 and the EMA factor $\alpha$ for teacher update is set to 0.999. The trainable parameters are all the parameters in ViT-Base.

**LAME[11] [3].** For fair comparison, we maintain a consistent batch size of 64 for LAME, aligning it with the same batch size used by other methods in our evaluation. We use the kNN affinity matrix with the value of k chosen from {1, 5, 10, 20}, and for all experiments, we consistently set it to 5 based on the optimal accuracy observed in Table 3.

**T3A[12] [21].** We follow all hyper-parameters that are set in T3A unless it does not provide. Specifically, the batch size is set to 64. The number of supports to restore $M$ is chosen from {1, 5, 20, 50, 100}, and for all experiments, we set it to 20 based on the optimal accuracy observed in Table 3.

**TPT[13] [47].** We follow all hyper-parameters that are set in TPT unless it does not provide. Specifically, we use AdamW as the update rule, with batch size of 1 and a learning rate of 0.005. Learnable tokens are initialized from the hard prompt of 'a photo of a'. The confidence threshold $\rho$ is set to 0.1 and the number of TPT steps is set to 1.

---

[7]https://github.com/mr-eggplant/SAR
[8]https://github.com/mr-eggplant/EATA
[9]https://github.com/Jhyun17/DeYO
[10]https://github.com/qinenergy/cotta
[11]https://github.com/fiveai/LAME
[12]https://github.com/matsuolab/T3A
[13]https://github.com/azshue/TPT

# D   More experimental results

In the main paper, we only report the results averaged over 15 corruptions in ImageNet-C (*i.e.*, in Table 1 and Table 4) or over each group of corruptions (*i.e.*, in Table 2) due to page limit. In this section, we offer additional results to enable a more comprehensive comparison.

**Results under lifelong test-time adaptation.** The knowledge sharing scheme in CoLA can also be used to mitigate catastrophic forgetting in lifelong TTA, similar to the aim of conventional supervised continual learning [68]. To further verify this, we provide more detailed results of Table 1, regarding the Accuracys on the first and last rounds of adaptation. From Table C, our CoLA outperforms the integrated baseline in the first round of adaptation, *e.g.*, the accuracy of 62.0% (ETA+CoLA) *vs.* 61.4% (ETA). This mainly stems from the latter corruptions encountered in the first round, where CoLA demonstrates superiority by leveraging the learned knowledge from previous adaptation with Eqn. (2). More importantly, our enhancement becomes particularly significant at the last round of adaptation, *e.g.*, the accuracy of 65.4% (ETA+CoLA) *vs.* 35.1% (ETA), further indicating our effectiveness in accumulating and utilizing learned knowledge for long-range adaptation. Note that CoLA applies weight decay on $\Delta\theta$ as claimed in Section C, thus achieving a lower performance on the first corruption. Here, the relatively limited performance of CoTTA [56] is attributed to its sensitivity to corruption order, as shown in Table D.

**Results under collaborative test-time adaptation.** We provide more detailed results of Table 2, regarding the Accuracys on each device. From Table D, our CoLA outperforms the integrated baseline from the adaptation to the second group of corruption, *e.g.*, the accuracy of 53.6% (SAR+CoLA) *vs.* 38.9% (SAR) on 'Gaussian' in Device 2. Moreover, this improvement becomes increasingly more pronounced as more knowledge is shared across devices, e.g., improving the accuracy from 37.5% (SAR) to 53.9% (SAR+CoLA) on 'Gaussian' in Device 3. This phenomenon underscores the importance of cross-device collaboration and our effectiveness regarding collaborative TTA.

**Results under single-domain test-time adaptation.** We provide more detailed results of Table 4, regarding the Accuracys under mild scenarios and online imbalanced label distribution shifts. From Table B, incorporating CoLA with existing TTA solutions enhances the adaptation performance on most corruptions under both scenarios, demonstrating our effectiveness. Specifically, CoLA showcases a more pronounced improvement in SAR and ETA. This can be attributed to the inadequacy of prediction entropy to identify reliable samples for model updates, thereby suffering more from error accumulation. CoLA alleviates this by dynamically favoring a more optimal checkpoint based on loss minimization instead of the newest saved one that may have learned from erroneous predictions, rendering CoLA more robust to noise. We also visualize $\alpha_i$ in Appendix E to offer more insights.

Table B: Comparisons on ImageNet-C (level 5) regarding **Accuracy (%)** under single-domain TTA. In mild scenarios, the test samples come in random order by following Tent [54]. ***Label Shifts*** is short for online imbalanced label distribution shifts, where test samples come in class order per SAR [39].

| *Mild Scenarios* | Gaus. | Shot | Imp. | Def. | Glass | Mot. | Zoom | Snow | Frost | Fog | Brit. | Contr. | Elas. | Pix. | JPEG | Avg. |
|---|---|---|---|---|---|---|---|---|---|---|---|---|---|---|---|---|
| NoAdapt | 9.5 | 6.7 | 8.2 | 29.0 | 23.4 | 33.9 | 27.1 | 15.9 | 26.5 | 47.2 | 54.7 | 44.1 | 30.5 | 44.5 | 47.8 | 29.9 |
| EATA [38] | 49.5 | 48.9 | 50.2 | 54.8 | 54.8 | 58.9 | 56.2 | 61.6 | 60.7 | 70.6 | 75.2 | 66.9 | 63.7 | 69.7 | 66.8 | 60.6 |
| SAR [39] | 44.0 | 24.2 | 45.3 | 53.0 | 49.9 | 55.7 | 51.0 | 57.4 | 45.1 | 66.6 | 74.7 | 64.5 | 55.3 | 66.7 | 64.0 | 54.5 |
| + CoLA (Ours) | 48.4 | 27.1 | 49.0 | 54.9 | 53.7 | 58.7 | 55.1 | 60.9 | 50.5 | 69.2 | 76.0 | 66.2 | 60.5 | 69.2 | 66.4 | 57.7 |
| ETA [38] | 51.9 | 51.9 | 52.8 | 57.6 | 57.6 | 62.2 | 60.0 | 66.1 | 65.1 | 72.5 | 77.4 | 67.6 | 66.1 | 72.0 | 69.2 | 63.3 |
| + CoLA (Ours) | 53.4 | 53.7 | 54.3 | 58.2 | 58.6 | 63.2 | 61.5 | 67.2 | 66.0 | 73.0 | 77.7 | 68.2 | 68.1 | 73.0 | 70.0 | 64.4 |
| DeYO [25] | 53.2 | 53.4 | 54.1 | 58.3 | 58.3 | 63.2 | 56.6 | 67.2 | 66.0 | 73.4 | 78.1 | 68.0 | 67.9 | 73.2 | 70.1 | 64.1 |
| + CoLA (Ours) | 54.3 | 50.6 | 55.0 | 58.4 | 59.4 | 64.2 | 60.1 | 68.1 | 66.5 | 73.9 | 78.1 | 68.3 | 68.8 | 73.9 | 70.4 | **64.7** |
| ***Label Shifts*** | Gaus. | Shot | Imp. | Def. | Glass | Mot. | Zoom | Snow | Frost | Fog | Brit. | Contr. | Elas. | Pix. | JPEG | Avg. |
| NoAdapt | 9.5 | 6.7 | 8.2 | 29.0 | 23.4 | 33.9 | 27.1 | 15.9 | 26.5 | 47.2 | 54.7 | 44.1 | 30.5 | 44.5 | 47.8 | 29.9 |
| EATA [38] | 36.1 | 35.7 | 35.4 | 45.0 | 42.8 | 52.0 | 45.1 | 55.0 | 48.7 | 62.1 | 73.0 | 42.9 | 55.8 | 63.9 | 62.7 | 50.4 |
| SAR [39] | 47.9 | 30.7 | 48.4 | 55.4 | 54.2 | 58.8 | 54.6 | 43.5 | 48.3 | 69.4 | 76.3 | 66.2 | 60.8 | 69.4 | 66.6 | 56.7 |
| + CoLA (Ours) | 49.6 | 50.4 | 50.7 | 56.4 | 56.4 | 60.3 | 57.3 | 40.9 | 36.6 | 71.5 | 77.2 | 67.0 | 64.2 | 71.3 | 68.5 | 58.5 |
| ETA [38] | 31.0 | 34.4 | 32.0 | 30.5 | 44.8 | 49.6 | 46.4 | 54.9 | 53.0 | 56.3 | 74.1 | 25.1 | 57.8 | 64.2 | 59.4 | 47.6 |
| + CoLA (Ours) | 40.2 | 37.6 | 42.4 | 47.1 | 49.6 | 52.5 | 53.0 | 59.7 | 58.4 | 65.0 | 74.6 | 50.9 | 61.3 | 68.9 | 66.3 | 55.2 |
| DeYO [25] | 53.0 | 53.9 | 54.5 | 57.8 | 59.0 | 63.9 | 12.7 | 68.0 | 66.1 | 73.2 | 77.9 | 66.6 | 68.9 | 73.7 | 70.6 | 61.3 |
| + CoLA (Ours) | 52.9 | 54.5 | 54.9 | 58.0 | 59.2 | 63.6 | 43.1 | 68.3 | 66.0 | 73.3 | 78.0 | 66.8 | 69.1 | 73.8 | 70.7 | **63.5** |

Table C: Comparison on ImageNet-C (severity level 5) regarding **Accuracy (%)** under lifelong adaptation for 10 rounds. Here, we provide additional results on the first and last rounds of adaptation.

| *Round 1* | Gaus. | Def. | Snow | Contr. | Shot | Glass | Frost | Elas. | Imp. | Mot. | Fog | Pix. | Brit. | Zoom | JPEG | Avg. |
|---|---|---|---|---|---|---|---|---|---|---|---|---|---|---|---|---|
| NoAdapt | 9.5 | 29.0 | 15.9 | 44.1 | 6.7 | 23.4 | 26.5 | 30.5 | 8.2 | 33.9 | 47.2 | 44.5 | 54.7 | 27.1 | 47.8 | 29.9 |
| CoTTA [56] | 24.4 | 35.2 | 34.6 | 48.3 | 41.9 | 40.0 | 52.6 | 49.8 | 47.0 | 44.5 | 48.8 | 54.6 | 63.7 | 35.7 | 53.1 | 44.9 |
| EATA [38] | 49.4 | 53.6 | 61.3 | 65.8 | 49.6 | 54.5 | 60.5 | 64.9 | 50.5 | 57.9 | 69.2 | 69.5 | 75.2 | 56.6 | 67.3 | 60.4 |
| SAR [39] | 44.0 | 51.9 | 58.5 | 63.3 | 48.8 | 53.0 | 61.0 | 63.5 | 51.2 | 57.2 | 66.9 | 69.0 | 76.5 | 54.0 | 67.2 | 59.1 |
| + CoLA (Ours) | 41.9 | 51.4 | 57.8 | 63.9 | 48.8 | 52.7 | 61.0 | 62.6 | 52.6 | 57.3 | 68.2 | 69.3 | 76.5 | 54.2 | 67.7 | 59.1 |
| ETA [38] | 51.9 | 55.6 | 64.5 | 64.6 | 53.2 | 55.5 | 62.4 | 66.4 | 52.1 | 57.9 | 67.5 | 69.8 | 75.3 | 57.7 | 66.2 | 61.4 |
| + CoLA (Ours) | 48.0 | 55.2 | 63.5 | 66.2 | 51.8 | 55.6 | 63.7 | 65.8 | 54.3 | 59.8 | 71.0 | 71.1 | 76.7 | 58.3 | 68.5 | **62.0** |
| DeYO [25] | 52.8 | 56.4 | 65.9 | 65.5 | 54.3 | 57.0 | 63.8 | 68.2 | 54.1 | 60.3 | 69.5 | 71.8 | 76.7 | 20.2 | 60.0 | 59.8 |
| + CoLA (Ours) | 49.7 | 55.9 | 65.0 | 66.4 | 53.2 | 56.9 | 65.0 | 67.4 | 55.2 | 61.4 | 72.0 | 72.3 | 77.7 | 37.8 | 69.7 | 61.7 |

| *Round 10* | Gaus. | Def. | Snow | Contr. | Shot | Glass | Frost | Elas. | Imp. | Mot. | Fog | Pix. | Brit. | Zoom | JPEG | Avg. |
|---|---|---|---|---|---|---|---|---|---|---|---|---|---|---|---|---|
| NoAdapt | 9.5 | 29.0 | 15.9 | 44.1 | 6.7 | 23.4 | 26.5 | 30.5 | 8.2 | 33.9 | 47.2 | 44.5 | 54.7 | 27.1 | 47.8 | 29.9 |
| CoTTA [56] | 22.1 | 20.7 | 27.2 | 22.4 | 25.2 | 22.0 | 29.6 | 29.4 | 25.8 | 24.4 | 28.3 | 31.3 | 38.2 | 20.2 | 31.5 | 26.5 |
| EATA [38] | 47.6 | 51.5 | 58.7 | 63.9 | 46.8 | 52.4 | 59.3 | 62.6 | 48.3 | 56.2 | 67.0 | 68.3 | 74.2 | 55.8 | 66.1 | 58.6 |
| SAR [39] | 51.3 | 55.4 | 62.5 | 63.3 | 52.7 | 55.9 | 62.0 | 64.3 | 53.2 | 59.1 | 67.8 | 70.4 | 75.9 | 55.9 | 67.1 | 61.1 |
| + CoLA (Ours) | 56.0 | 58.4 | 67.8 | 67.8 | 57.2 | 59.2 | 66.2 | 69.4 | 57.4 | 63.7 | 72.7 | 73.3 | 77.8 | 63.1 | 70.4 | 65.4 |
| ETA [38] | 32.6 | 28.8 | 33.4 | 17.5 | 31.4 | 31.9 | 36.8 | 42.3 | 32.1 | 29.2 | 30.6 | 46.0 | 57.7 | 34.9 | 41.7 | 35.1 |
| + CoLA (Ours) | 55.8 | 58.3 | 68.4 | 67.9 | 57.1 | 59.2 | 66.3 | 69.7 | 56.9 | 64.2 | 73.1 | 73.2 | 77.5 | 64.2 | 69.8 | 65.4 |
| DeYO [25] | 0.1 | 0.1 | 0.1 | 0.1 | 0.1 | 0.1 | 0.1 | 0.1 | 0.1 | 0.1 | 0.1 | 0.1 | 0.1 | 0.1 | 0.1 | 0.1 |
| + CoLA (Ours) | 56.9 | 58.9 | 69.7 | 68.2 | 58.1 | 59.8 | 66.9 | 70.8 | 58.2 | 64.7 | 73.7 | 73.9 | 78.0 | 53.9 | 70.9 | **65.5** |

Table D: Effectiveness under collaborative adaptation across **resource-abundant principal devices** w.r.t. **Acc. (%)**. Results are evaluated on ImageNet-C (severity level 5, containing 15 corruption types of 4 groups). We share learned weights across devices post-adaptation to each group of corruptions.

| *Device 1* | Gaus. | Shot | Imp. | Def. | Glass | Mot. | Zoom | Snow | Frost | Fog | Brit. | Contr. | Elas. | Pix. | JPEG | Avg. |
|---|---|---|---|---|---|---|---|---|---|---|---|---|---|---|---|---|
| NoAdapt | 9.5 | 6.7 | 8.2 | 29.0 | 23.4 | 33.9 | 27.1 | 15.9 | 26.5 | 47.2 | 54.7 | 44.1 | 30.5 | 44.5 | 47.8 | 29.9 |
| CoTTA [56] | 17.7 | 28.4 | 40.5 | 37.0 | 43.4 | 46.5 | 38.1 | 39.1 | 49.1 | 48.6 | 63.8 | 41.5 | 42.5 | 54.0 | 52.2 | 42.8 |
| EATA [38] | 50.4 | 54.3 | 55.7 | 53.8 | 56.1 | 59.6 | 58.6 | 62.3 | 63.8 | 70.5 | 75.9 | 66.2 | 64.1 | 70.1 | 68.3 | 62.0 |
| SAR [39] | 44.5 | 51.8 | 54.8 | 51.3 | 53.6 | 57.7 | 55.0 | 59.7 | 62.0 | 67.2 | 76.3 | 63.1 | 59.4 | 68.5 | 67.2 | 59.5 |
| + CoLA (Ours) | 44.5 | 51.8 | 54.8 | 56.4 | 56.5 | 61.2 | 57.9 | 64.7 | 64.4 | 71.7 | 76.8 | 66.4 | 66.7 | 72.2 | 69.6 | 62.4 |
| ETA [38] | 51.9 | 56.3 | 57.2 | 53.1 | 56.7 | 58.8 | 58.9 | 62.2 | 63.3 | 68.9 | 75.5 | 62.7 | 64.3 | 69.9 | 67.1 | 61.8 |
| + CoLA (Ours) | 52.1 | 56.3 | 57.1 | 57.2 | 58.0 | 62.1 | 62.6 | 67.7 | 66.3 | 72.7 | 77.1 | 66.0 | 69.0 | 72.5 | 69.7 | **64.4** |
| DeYO [25] | 52.9 | 57.7 | 58.3 | 54.9 | 58.0 | 61.4 | 25.5 | 61.5 | 63.9 | 70.1 | 77.0 | 63.5 | 67.0 | 71.6 | 69.2 | 60.8 |
| + CoLA (Ours) | 52.9 | 57.6 | 58.2 | 57.9 | 58.3 | 62.5 | 41.6 | 67.8 | 66.4 | 73.0 | 77.5 | 66.4 | 70.5 | 73.3 | 70.5 | 63.6 |

| *Device 2* | Def. | Glass | Mot. | Zoom | Gaus. | Shot | Imp. | Contr. | Elas. | Pix. | JPEG | Snow | Frost | Fog | Brit. | Avg. |
|---|---|---|---|---|---|---|---|---|---|---|---|---|---|---|---|---|
| NoAdapt | 29.0 | 23.4 | 33.9 | 27.1 | 9.5 | 6.7 | 8.2 | 44.1 | 30.5 | 44.5 | 47.8 | 15.9 | 26.5 | 47.2 | 54.7 | 29.9 |
| CoTTA [56] | 31.4 | 31.7 | 43.9 | 38.1 | 27.0 | 37.7 | 46.7 | 45.2 | 46.3 | 56.2 | 54.0 | 47.0 | 50.9 | 49.3 | 63.5 | 44.6 |
| EATA [38] | 55.4 | 57.2 | 60.5 | 59.5 | 48.5 | 53.1 | 55.0 | 65.0 | 64.0 | 70.5 | 68.4 | 62.9 | 64.0 | 71.0 | 76.1 | 62.1 |
| SAR [39] | 53.0 | 53.4 | 58.7 | 55.3 | 38.9 | 51.5 | 54.7 | 62.2 | 57.6 | 68.2 | 68.0 | 59.9 | 61.9 | 67.6 | 76.3 | 59.1 |
| + CoLA (Ours) | 52.9 | 53.4 | 58.6 | 55.1 | 53.6 | 55.4 | 56.1 | 66.4 | 61.9 | 71.1 | 69.2 | 66.2 | 65.6 | 72.9 | 77.3 | 62.4 |
| ETA [38] | 57.6 | 58.8 | 61.7 | 61.1 | 47.4 | 53.2 | 54.4 | 57.3 | 64.6 | 70.5 | 67.7 | 62.1 | 62.7 | 69.0 | 75.6 | 61.6 |
| + CoLA (Ours) | 57.5 | 58.3 | 61.2 | 61.0 | 54.8 | 57.0 | 57.1 | 64.6 | 68.4 | 72.2 | 70.1 | 67.5 | 66.0 | 72.8 | 77.3 | **64.4** |
| DeYO [25] | 58.1 | 59.5 | 63.0 | 41.8 | 39.3 | 50.5 | 50.4 | 61.4 | 66.8 | 71.6 | 68.9 | 64.6 | 63.9 | 70.5 | 76.9 | 60.5 |
| + CoLA (Ours) | 58.3 | 59.1 | 62.6 | 39.3 | 47.9 | 57.5 | 58.2 | 66.6 | 69.9 | 73.0 | 70.5 | 68.3 | 66.9 | 73.0 | 77.7 | 63.3 |

| *Device 3* | Snow | Frost | Fog | Brit. | Contr | Elas. | Pix. | JPEG | Def. | Glass | Mot. | Zoom | Gaus. | Shot | Imp. | Avg. |
|---|---|---|---|---|---|---|---|---|---|---|---|---|---|---|---|---|
| NoAdapt | 15.9 | 26.5 | 47.2 | 54.7 | 44.1 | 30.5 | 44.5 | 47.8 | 29.0 | 23.4 | 33.9 | 27.1 | 9.5 | 6.7 | 8.2 | 29.9 |
| CoTTA [56] | 26.1 | 48.0 | 56.5 | 71.2 | 53.1 | 45.3 | 60.9 | 60.6 | 42.3 | 42.8 | 46.8 | 38.2 | 30.2 | 39.8 | 46.2 | 47.2 |
| EATA [38] | 63.8 | 65.3 | 71.8 | 76.6 | 66.9 | 64.7 | 70.5 | 68.8 | 55.7 | 56.7 | 60.2 | 59.1 | 47.5 | 52.7 | 54.3 | 62.3 |
| SAR [39] | 57.9 | 63.1 | 68.6 | 76.3 | 64.6 | 58.0 | 67.2 | 67.7 | 54.5 | 54.3 | 58.3 | 54.6 | 37.5 | 51.0 | 54.6 | 59.2 |
| + CoLA (Ours) | 57.9 | 63.0 | 68.0 | 76.2 | 64.6 | 59.7 | 69.1 | 67.8 | 56.5 | 57.9 | 61.6 | 59.2 | 53.9 | 56.3 | 56.5 | 61.9 |
| ETA [38] | 66.0 | 66.3 | 71.9 | 77.0 | 66.0 | 65.6 | 70.7 | 68.8 | 55.3 | 56.8 | 60.1 | 59.9 | 44.3 | 50.9 | 52.9 | 62.2 |
| + CoLA (Ours) | 66.0 | 66.0 | 71.9 | 76.9 | 65.4 | 66.4 | 71.6 | 69.1 | 56.7 | 58.0 | 61.8 | 62.1 | 54.0 | 56.0 | 55.9 | **63.9** |
| DeYO [25] | 67.2 | 66.9 | 72.7 | 77.7 | 66.3 | 67.4 | 71.9 | 69.6 | 56.6 | 58.1 | 61.6 | 28.2 | 11.7 | 1.0 | 0.1 | 51.8 |
| + CoLA (Ours) | 67.0 | 66.6 | 72.7 | 77.6 | 66.1 | 67.6 | 72.5 | 69.7 | 56.5 | 57.5 | 61.4 | 39.6 | 49.1 | 56.9 | 56.8 | 62.5 |

# E   Additional discussions

**Robustness of CoLA against potential harmful knowledge.** Note that CoLA conducts test-time learning to adaptively aggregate prior knowledge. If the prior knowledge shared from some devices is harmful, using such knowledge shall not decrease the test-time objective. Thus, such prior knowledge will not be used for model adaptation. In this sense, CoLA remains stable even with harmful prior knowledge (*e.g.*, the domain knowledge in some devices is not helpful for principal agents to do TTA). This stability also benefits from our initialization strategy as shown in Table E, which carefully initializes $\alpha$ and $\Delta\theta$ to prevent the aggregation of potentially harmful knowledge for stable warm-up.

Table E: Robustness of our CoLA using only pre-trained parameters and $N$ harmful prior knowledge, *i.e.*, the randomly initialized domain vectors. Results are obtained on ImageNet-C (Gaussian, level 5).

| Method | $N = 0$ | $N = 2$ | $N = 4$ | $N = 100$ | $N = 1,000$ |
|---|---|---|---|---|---|
| ETA+CoLA (Equal $\alpha$) | 51.97 | 0.10 | 0.10 | 0.10 | 0.10 |
| ETA+CoLA (Random $\alpha$) | 51.97 | 10.46 | 0.15 | 0.10 | 0.10 |
| ETA+CoLA (Ours) | 51.97 | 52.02 | 52.04 | 52.04 | 52.04 |

**Scalability of CoLA with more collaborative devices.** CoLA scales well with an increasing number of devices, *i.e.*, with more shared domain vectors. From Table F, ETA/ETA+CoLA consistently benefits from additional participating devices, *e.g.*, ETA+CoLA achieves an accuracy of 65.2% with 11 devices compared to 61.8% with one device. This highlights the importance of cross-device collaboration and CoLA 's effectiveness under more large-scale multi-device collaborative TTA.

Table F: Effectiveness of CoLA with an increasing number of principal devices (with back-propagation capability). Here, each device continuously encounters 15 domains from ImageNet-C (level 5) in different domain orders. Results are averaged over all principal devices.

| #Devices | 1 | 3 | 5 | 7 | 9 | 11 |
|---|---|---|---|---|---|---|
| SAR+CoLA | 59.7 | 62.1 | 62.8 | 63.6 | 63.8 | **63.9**$_{(+4.2)}$ |
| ETA+CoLA | 61.8 | 63.9 | 64.3 | 64.9 | 65.0 | **65.2**$_{(+3.4)}$ |

**Efficiency of CoLA with increasing domain vectors.** Our CoLA is both computation and memory efficient at exploiting domain vectors. From Table H, CoLA efficiently scales to over 10,000 domain vectors, incurring only an additional 11s of runtime and 1,502MB of extra memory, yet remains substantially more efficient than CoTTA. We believe that 10,000 domain vectors should be adequate for handling most real-world applications with proper management.

Table G: Efficiency comparison on ImageNet-C (Gaussian, level 5) using a single A100. $N$ is the number of domain vectors, which we initialize as random parameters in this table.

| Method | ETA | +CoLA ($N = 1$) | +CoLA ($N = 100$) | +CoLA ($N = 10,000$) | CoTTA |
|---|---|---|---|---|---|
| Time (s) | 109 | 110 | 112 | 120 | 937 |
| Memory (MB) | 7,433.2 | 7,433.7 | 7,448.2 | 8,935.5 | 21,628.6 |

**Sensitivity of threshold $z$ for shift detection.** CoLA remains effective among a wide range of threshold $z$, as shown in Table A. From the results, while a stricter threshold saves more domain vectors, CoLA achieves a stable performance of around 64.7%. When threshold $z$ increases, CoLA saves significantly fewer domain vectors and still enhances the performance significantly, *i.e.*, the average accuracy of 62.2% in ETA+CoLA ($z$=10) *vs.* 46.4% in ETA.

Table H: Sensitivity of threshold $z$. Experiments follow the settings of Table 1, *i.e.*, single-device lifelong adaptation, and CoLA is incorporated with ETA. We report average accuracy over 10 rounds, each comprising 15 corruptions of ImageNet-C. The average accuracy of the ETA baseline is 46.4%.

| | CoLA ($z$=0.01) | CoLA ($z$=0.05) | CoLA ($z$=0.1) | CoLA ($z$=1) | CoLA ($z$=10) |
|---|---|---|---|---|---|
| Avg. Acc. | 64.7 | 64.6 | 64.8 | 63.8 | 62.2 |
| #Saved Vectors | 20740 | 369 | 169 | 110 | 48 |

**Advantages of CoLA over FedAvg [32] in collaborative TTA.** We further compare our CoLA with FedAvg [32] in cross-device collaborative/federated learning. From Table I, CoLA consistently outperforms FedAvg when incorporated with the baseline for collaborative TTA. More importantly, FedAvg, which simply averages the model parameters on different devices, may deteriorate model performance (*i.e.*, the average accuracy of 61.2% in ETA *vs.* 58.0% in ETA+FedAvg). This is because different devices may encounter different distribution shifts, and thus the knowledge from other devices may not be beneficial for adapting to the current domain. In contrast, our CoLA addresses this by learning at test time to optimize the aggregation of knowledge from different devices.

Table I: Effectiveness of CoLA and FedAvg [32] for cross-device collaborative TTA w.r.t. **Acc. (%)**. The experiments follow the settings of Table 2 in the main paper.

| Method | Device 1 (Adapt →) | | | | Device 2 (Adapt →) | | | | Device 3 (Adapt →) | | | | Avg. |
|---|---|---|---|---|---|---|---|---|---|---|---|---|---|
| | Noise | Blur | Weat. | Digit. | Blur | Noise | Digit. | Weat. | Weat. | Digit. | Blur | Noise | |
| NoAdapt | 8.2 | 28.4 | 36.1 | 41.7 | 28.4 | 8.2 | 41.7 | 36.1 | 36.1 | 41.7 | 28.4 | 8.2 | 28.6 |
| EATA [38] | 53.5 | 57.0 | 68.1 | 67.2 | 58.1 | 52.2 | 67.0 | 68.5 | 69.4 | 67.7 | 57.9 | 51.5 | 61.5 |
| SAR [39] | 50.4 | 54.4 | 66.3 | 64.5 | 55.1 | 48.3 | 64.0 | 66.4 | 66.5 | 64.3 | 55.4 | 47.7 | 58.6 |
| + FedAvg [32] | 50.4 | 56.6 | 67.6 | 66.1 | 55.1 | 52.2 | 65.6 | 68.1 | 66.3 | 64.9 | 57.2 | 51.8 | 60.2 |
| + CoLA (Ours) | 50.4 | 58.0 | 69.4 | 68.7 | 55.0 | 55.0 | 67.1 | 70.5 | 66.3 | 65.3 | 58.8 | 55.5 | 61.7 |
| ETA [38] | 55.2 | 56.9 | 67.5 | 66.0 | 59.8 | 51.7 | 65.0 | 67.4 | 70.3 | 67.8 | 58.0 | 49.4 | 61.2 |
| + FedAvg [32] | 55.2 | 58.9 | 65.1 | 62.5 | 59.6 | 51.0 | 63.4 | 63.7 | 70.3 | 46.2 | 54.0 | 46.4 | 58.0 |
| + CoLA (Ours) | 55.2 | 60.0 | 70.9 | 69.3 | 59.5 | 56.3 | 68.8 | 70.9 | 70.2 | 68.1 | 59.7 | 55.3 | **63.7** |

**Robustness of CoLA under small batch sizes.** The stability of CoLA under small batch sizes is primarily determined by the base algorithms, such as ETA and SAR, rather than CoLA itself. This is because CoLA is a plug-and-play module designed to be incorporated with existing methods. We provide further empirical results to verify this robustness. From Table J, CoLA achieves a consistent result when incorporated with SAR, demonstrating no performance degradation as the batch size reduces from 16 to 2. Meanwhile, CoLA can also help improve stability under small batch sizes. For instance, as the batch size reduces from 4 to 2, ETA+CoLA achieves nearly no performance degradation while the baseline ETA's performance degrades by 1.4%.

Table J: Robustness of our CoLA under various batch sizes. We follow the same settings of Table 2 in the main paper and report **average accuracy** over all devices and corruptions here.

| Method | $BS = 64$ | $BS = 16$ | $BS = 4$ | $BS = 2$ |
|---|---|---|---|---|
| SAR [39] | 58.6 | 58.8 | 58.9 | 58.7 |
| + CoLA (Ours) | 61.7 | 61.6 | 61.7 | 61.7 |
| ETA | 61.2 | 60.5 | 58.5 | 57.1 |
| +CoLA (Ours) | 63.7 | 62.8 | 61.7 | 61.5 |

**Effectiveness of CoLA on ResNet models.** Table K demonstrates the effectiveness of our CoLA on ResNet-50, where CoLA updates and stores the affine parameters of batch normalization layers in ResNet. From Table K, CoLA consistently enhances the performance of ETA/EATA/SAR throughout 10 rounds of adaptation and addresses the issue of performance degradation in long-term adaptation. These results are consistent with Table 1 in the main paper using ViT-Base, further indicating CoLA's effectiveness in accumulating and exploiting learned knowledge with diverse model architectures.

Table K: Effectiveness of CoLA on ResNet-50 in the lifelong TTA scenarios following Table 1.

| Round | Time: | | | | | | | | | | Average |
|---|---|---|---|---|---|---|---|---|---|---|---|
| | 1 | 2 | 3 | 4 | 5 | 6 | 7 | 8 | 9 | 10 | |
| NoAdapt | 18.0 | 18.0 | 18.0 | 18.0 | 18.0 | 18.0 | 18.0 | 18.0 | 18.0 | 18.0 | 18.0 |
| CoTTA [56] | 33.4 | 23.6 | 9.6 | 2.2 | 1.2 | 1.2 | 1.2 | 1.2 | 1.2 | 1.2 | 7.6 |
| SAR [39] | 35.9 | 17.5 | 21 | 36.2 | 18 | 13.6 | 35.8 | 16.5 | 13.5 | 36.1 | 24.4 |
| + CoLA (Ours) | 39.4 | 42 | 43 | 43.5 | 43.9 | 44.4 | 44.7 | 45 | 45.2 | 45.3 | $43.6_{(+19.2)}$ |
| ETA [38] | 42.4 | 40.4 | 38.9 | 37.6 | 37 | 36 | 35.9 | 35.2 | 35.1 | 34.7 | 37.3 |
| + CoLA (Ours) | 46.5 | 46.5 | 49.5 | 49.5 | 49.7 | 49.8 | 49.9 | 49.9 | 49.9 | 49.8 | $49.3_{(+12.0)}$ |
| EATA [38] | 47.5 | 47.3 | 47.1 | 46.7 | 46.8 | 46.6 | 46.4 | 46.3 | 46.2 | 46.2 | 46.7 |
| + CoLA (Ours) | 48.2 | 49.5 | 50.0 | 50.1 | 50.2 | 50.2 | 50.3 | 50.3 | 50.3 | 50.2 | $\mathbf{49.9}_{(+3.2)}$ |

**Effectiveness of CoLA in sample efficiency on unseen distributions.** We validate the effectiveness of Eqn. (2) to facilitate sample-efficient TTA on unseen distributions. From Figure B, CoLA consistently enhances the sample efficiency and the overall performance, *i.e.*, with an up to $30.0\times$ speed up on both ImageNet-R and ImageNet-Sketch, indicating that the effectiveness of CoLA is not limited to previously encountered distributions. More interestingly, compared with SAR, CoLA helps mitigate overfitting on ImageNet-Sketch. This can be attributed to that, in the unsupervised adaptation, SAR learns on more erroneous predictions since its performance is particularly limited at the beginning of adaptation. This phenomenon further demonstrates the importance of sample efficiency and our effectiveness in leveraging shared knowledge for efficient TTA.

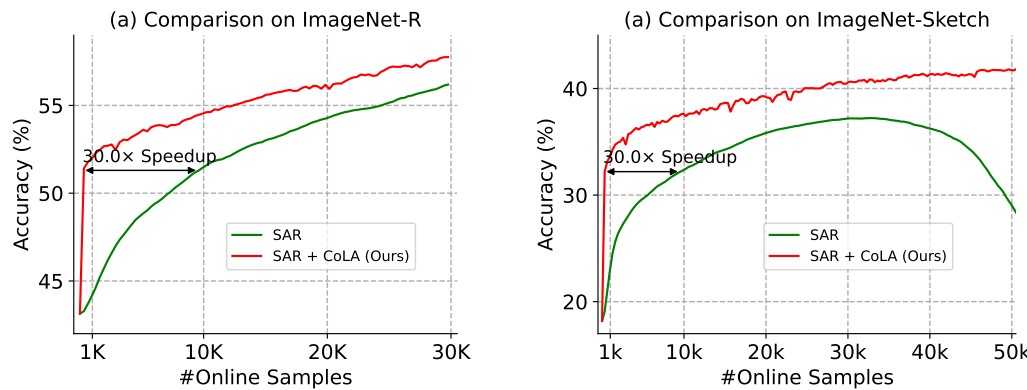

Figure B: Additional comparisons w.r.t sample efficiency on **unseen distributions**. Here, CoLA leverages learned weights on ImageNet-C (*i.e.*, from SAR+CoLA in Table 2, the fifth row) while the effectiveness in sample efficiency is evaluated on ImageNet-R and ImageNet-Sketch.

**Effectiveness of CoLA in mitigating error accumulation.** We provide visualization of the learned $\beta_i$ in Figure C. Here, all domain vectors are learned on the same domain, *i.e.*, Gaussian Noise, where we save learned weights post-adaptation to 10 batches of samples. Generally, a later saved domain vector should be prioritized as it learns on more samples. Nevertheless, in the context of unsupervised TTA, a model may learn from erroneous pseudo-labels, where the performance would significantly deteriorate, known as error accumulation. From Figure C, CoLA adaptively assigns lower $\beta_i$ on domain vectors that may have accumulated error, *e.g.*, from the 80-th to the 90-th domain vectors, according to loss optimization. Thus, CoLA demonstrates potential effectiveness in mitigating error accumulation, enhancing ETA's performance by +7.6% on ImageNet-C under label distribution shifts.

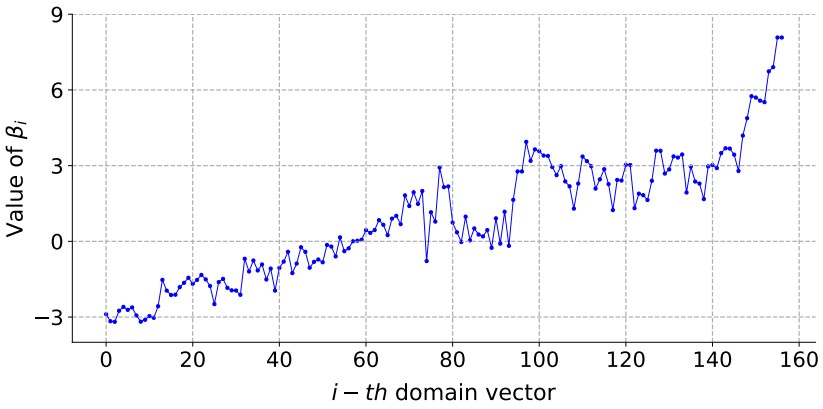

Figure C: Visualization of $\beta_i$ for ETA+CoLA on ImageNet-C(severity level 5, Gaussian) under online imbalanced label distribution shifts. CoLA saves learned weights for every adaptation to 10 batches of samples while no weights are discarded for visualization. Learned temperature $T_l$ is 0.49.

**Effectiveness of our domain shift detector.** We provide additional results to demonstrate the effectiveness of our domain distance function, *i.e.*, Eqn. (7), for shift detection. In our experiments,

we consistently set $\phi_d$ to 0.1 for shift detection. From Figure D, our domain shift detector demonstrates sensitivity between corruptions from different groups, *e.g.*, with a distance of 99 between 'impulse noise' and 'contrast', and a distance of 32 between 'gaussian noise' and 'defocus blur'.

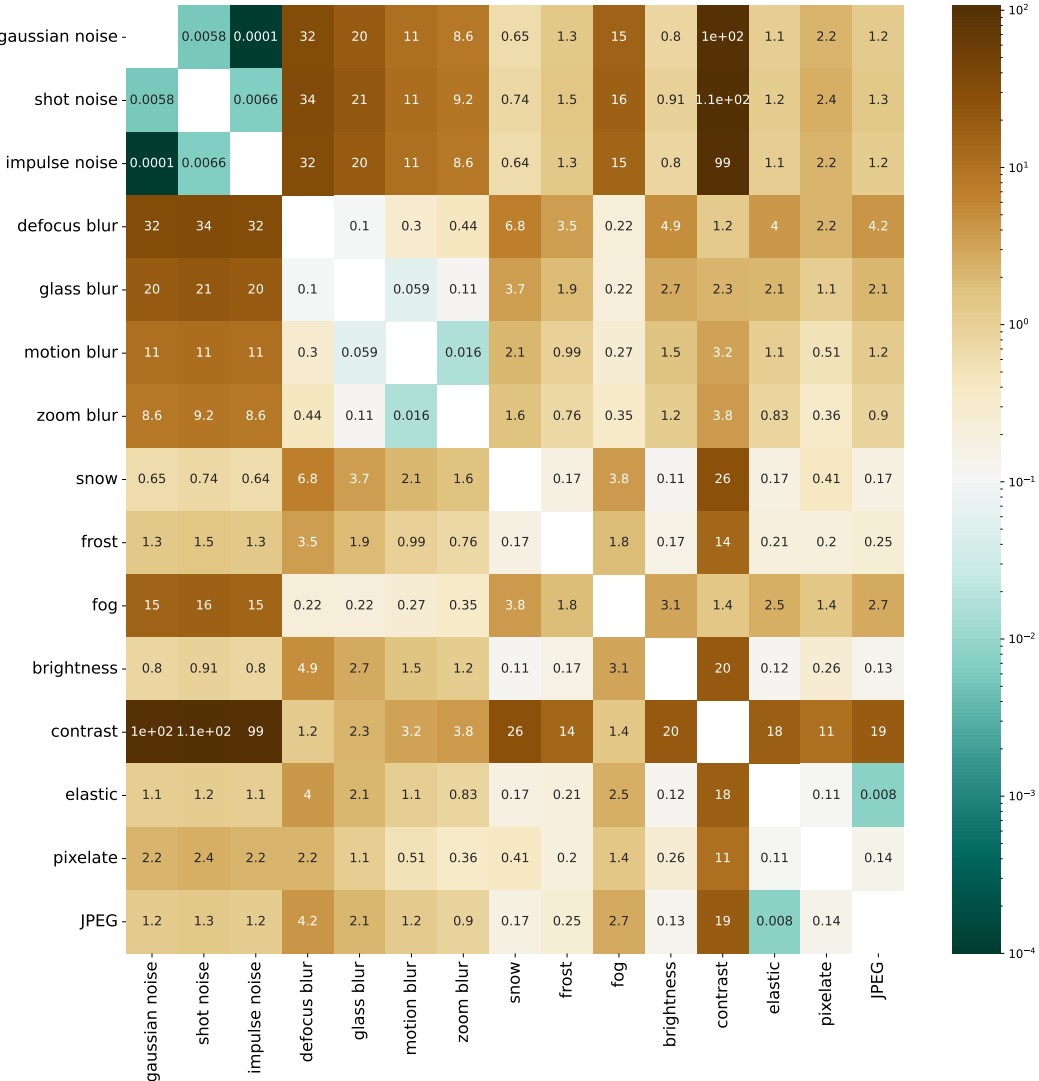

Figure D: Effectiveness of our domain distance function, *i.e.*, $D(\cdot, \cdot)$ defined in Eqn. (7), to capture the magnitude of domain shift. Here, we estimate the domain distance between each corruption in ImageNet-C (level 5, with 15 corruptions). The statistic of each domain is estimated via Eqn. (6).

**Statistical comparison.** We re-run Table 2 in the main paper with 5 different seeds and report the mean and std of each method in Table L. The results show that CoLA performs stably with small stds and it lowers the std of ETA&DeYo, suggesting CoLA's stability.

Table L: Statistical comparison. Experiments follow the settings of Table 2.

| Method | Device 1 (Adapt $\rightarrow$) | | | | Device 2 (Adapt $\rightarrow$) | | | | Device 3 (Adapt $\rightarrow$) | | | | Avg. |
|---|---|---|---|---|---|---|---|---|---|---|---|---|---|
| | Noise | Blur | Weat. | Digit. | Blur | Noise | Digit. | Weat. | Weat. | Digit. | Blur | Noise | |
| ETA [38] | 55.1 | 56.8 | 67.3 | 58.5 | 59.7 | 51.6 | 52.2 | 66.0 | 70.4 | 67.8 | 57.9 | 49.5 | $60.2_{\pm 1.8}$ |
| + CoLA (Ours) | 55.1 | 59.7 | 70.3 | 68.7 | 59.5 | 56.3 | 65.2 | 70.3 | 70.3 | 68.0 | 59.2 | 54.6 | $63.1_{\pm 0.7}$ |
| SAR [39] | 50.3 | 54.4 | 66.3 | 64.7 | 55.1 | 48.0 | 63.9 | 66.5 | 66.5 | 64.4 | 55.6 | 47.0 | $58.6_{\pm 0.1}$ |
| + CoLA (Ours) | 50.3 | 57.9 | 69.1 | 68.1 | 55.0 | 54.8 | 67.2 | 70.1 | 66.4 | 65.1 | 58.7 | 54.5 | $61.4_{\pm 0.3}$ |
| DeYO [25] | 56.3 | 50.5 | 67.6 | 67.9 | 55.4 | 44.9 | 53.5 | 55.4 | 71.1 | 68.9 | 52.0 | 26.6 | $55.8_{\pm 3.9}$ |
| + CoLA (Ours) | 56.2 | 54.7 | 70.4 | 69.2 | 54.1 | 55.9 | 68.8 | 70.6 | 71.1 | 68.7 | 52.4 | 54.2 | $62.2_{\pm 0.3}$ |

# F    Limitations and future works

**Finetuning LLM.** We mainly verify our CoLA with vision models in the context of test-time adaptation. Since our formulation in Eqn. (2) does not necessitate being optimized during testing, it's an interesting future work to verify CoLA on more scenarios. For instance, finetuning the large language models, where multiple finetuned weights, *e.g.*, using LoRA [20], are publicly available.

**Shrinking the shared vectors.** Our CoLA continuously expands the size of the shared domain vectors across devices. Intuitively, on the same domain, one can reduce memory consumption by preserving only the best-performing domain vector. Although we have demonstrated a feasible strategy for shared vectors shrinking on the single-domain adaptation, where we discard the unused ones according to $\alpha_i$ in Table 4. Nevertheless, it's still challenging to perform share vectors shrinking across multiple devices and we leave it for future works.

# G    Broader impact

This paper aims to advance the field of test-time adaptation for out-of-distribution generalization. The societal impact of our work lies primarily in its potential to expand the usability of machine learning models in real-world settings, particularly on self-driving cars, embodied agents/robots, *etc*. By enhancing the performance of machine learning models on various real-world devices, our method helps make AI technology more broadly accessible. Ethically, our approach eliminates the need for data transfer between devices, thereby improving data privacy and security.

