# OpenReview forum: "Cross-Device Collaborative Test-Time Adaptation"
_NeurIPS.cc/2024/Conference — NeurIPS 2024 poster_

### Official Review · Reviewer_L5Gu · 2024-06-28

**Soundness:** 3
**Presentation:** 2
**Contribution:** 3
**Rating:** 5
**Confidence:** 4

**Summary:**

This paper proposes a novel Collaborative Lifelong Adaptation (CoLA) method for test-time adaptation (TTA) across multiple devices. The key idea is to accumulate and share knowledge learned during adaptation across devices, enabling more efficient and effective adaptation to distribution shifts at test time. The authors evaluate CoLA in various scenarios, showcasing the effectiveness of the plug-and-play CoLA method.

**Strengths:**

1. The authors propose a new and novel setting of cross-device collaborative TTA.
1. The authors conducted intensive experiments with transformer-based models on various scenarios.

**Weaknesses:**

Although the paper proposes an innovative research direction, I have concerns about its positioning and presentation of the results.

1. **Limited comparisons with test-time federated learning [1, 2]**. Collaborative TTA could be considered as a variation of federated learning. Please thoroughly discuss the relation with federated learning. The paper should thoroughly discuss this relationship and provide comparative experimental results if possible.
1. **Insufficient scalability analysis**. The paper does not address how the method scales with an increasing number of devices and domains, particularly regarding the growth of the shared domain vector set T.
1. **Vague approach to lifelong adaptation**. While the paper claims to address lifelong adaptation, it doesn't fully explain how the method performs in lifelong scenarios. It's unclear how the proposed methods tackle potential performance or memory issues arising from accumulating domain vectors over time.
1. **Incomplete experimental details**:
   - Lack of explanation on the detailed experiment settings, including the details of resource-abundant and resource-limited devices.
   - The authors only applied CoLA on ETA, SAR, and DeYO. Please discuss why these specific methods were chosen and whether they are applicable to other TTA methods.
   - The experiment results use different baselines on each table; please elaborate on the baseline selection criteria.
1. **Lack of statistical significance**. The paper does not report any error bars of the experimental results.



[1] Bao, Wenxuan, et al. "Adaptive test-time personalization for federated learning." Advances in Neural Information Processing Systems 36 (2024).

[2] Jiang, Liangze, and Tao Lin. "Test-Time Robust Personalization for Federated Learning." The Eleventh International Conference on Learning Representations.

**Questions:**

1. Please explain details on the limitations above.
1. What are the overheads of the shared domain vectors? Can you provide any numerical example of the growth of shared vectors?
1. Given that CoLA involves optimizing both $\alpha$ and $\Delta \theta$, how does the method achieve such minimal computational overhead (only a 3-second increase) compared to the baseline ETA, as reported in Table 5?

**Important**: I will raise the ratings if concerns are resolved.

**Limitations:**

The paper properly discussed the limitations and societal impacts.
Still, it's worth mentioning the limited applicability and scalability of continuously stacking shared domain vectors, which is discussed by the authors.

---

> ### Author Rebuttal · Authors · 2024-08-07
>
> > Q1. Comparisons with test-time federated learning.
>
> A. Please refer to our General Response.
>
> > Q2. More scalability analysis regarding CoLA.
>
> A. CoLA scales well with an increasing number of devices, i.e., with more shared domain vectors. From Table A, ETA/SAR+CoLA consistently benefits from additional participating devices, e.g., ETA+CoLA achieves an accuracy of 65.2% with 11 devices compared to 61.8% with just one device. This highlights the importance of cross-device collaboration and demonstrates CoLA’s effectiveness under more large-scale multi-device collaborative TTA.
>
>
> Table A. Effectiveness of CoLA with an increasing number of principal devices (with back-propagation capability). Here, each device continuously encounters 15 domains from ImageNet-C (level 5) in different domain orders.
> |#Devices|1|3|5|7|9|11|
> |-|:-:|:-:|:-:|:-:|:-:|:-:|
> |SAR+CoLA|59.7|62.1|62.8|63.6|63.8|63.9|
> |ETA+CoLA|61.8|63.9|64.3|64.9|65.0|65.2|
>
>
>
> > Q3. More clarifications of CoLA for lifelong adaptation and its scalability with accumulating domain vectors.
>
> A. For lifelong adaptation, CoLA automatically detects domain changes and then explicitly saves each domain vector once domian is changed for future reuse. When similar domain data reappears, CoLA can directly reuse the previously encountered knowledge through reprogramming, i.e., learning a weight $\alpha$ to reweight existing domain vectors while simultaneously learning new domain vectors through gradient optimization, thereby overcoming catastrophic forgetting.
>
> According to Q2, Reviewer sr9b's Q6, and Reviewer SsoW's Q3-2, CoLA scales well in terms of both performance and efficiency with accumulating domain vectors.
>
>
> > Q4-1. Lack of explanation on the detailed experiment settings, including the details of resource-abundant and resource-limited devices.
>
> A. We have provided detailed settings for each experiments in Appendix D.2 and do not include them in the main paper due to page limits. We will make this clearer in the main paper of the revision.
>
> For resource-abundant/limited devices, we term devices with backpropagation capacity, like GPU servers, as resource-abundant. Devices without backpropagation capacity or requiring very low latency, such as smartphones and surveillance cameras, are considered resource-limited. However, in our experiments, we simulate these devices using GPUs, i.e., run all experiments on GPUs, which does not affect the fairness of comparisons.
>
>
>
>
>
> > Q4-2. The authors only applied CoLA on ETA, SAR, and DeYO. Please discuss why these specific methods were chosen and whether they are applicable to other TTA methods.
>
> A. We select ETA [ICML 2022] and SAR [ICLR 2023 Oral] as they are strong baselines on various benchmarks and widely used by existing TTA literature. Moreover, DeYO, accepted as ICLR 2024 spotlight, is one of the most advanced TTA method.
>
> CoLA is applicable but not limited to the above methods. This is because compared to conventional TTA objective, CoLA introduces only an adaptive aggregation term in optimization, $\sum\alpha_i\Delta\theta_i$, where $\Delta\theta_i$ are frozen and only $\alpha_i$ are learnable. **We do not limit the loss funtion to optimze these $\alpha_i$.** Thus, CoLA can integrate and benefit from the loss designs of the advanced TTA solutions.
>
> Please also refer to Reviewer 1r2f's Q3 in which CoLA works well with EATA.
>
>
> > Q4-3. The experiment results use different baselines on each table; please elaborate on the baseline selection criteria.
>
> A. We selected baselines based on whether the method relies on backpropagation, aligning with the computational resources of each device.
>
> In Tables 1-2 of the main paper, experiments are conducted on principal agents (with backpropagation capability), and thus the baselines are all backpropagation-based. In Table 3 of the main paper, the experiments are conducted on resource-limited devices (follower agents), in which backpropagation-based methods are infeasible. Therefore, all baselines in Table 3 are forward-only methods.
>
>
> > Q5. Error bars of the experimental results.
>
>
> A. We re-run Table 2 in the main paper with 5 different seeds and report the mean and std of each method in Table B. The results show that CoLA performs stably with small stds and it lowers the std of ETA\&DeYo by alleviating the model collapse issue during a long-term adaptation process, suggesting CoLA's stability.
>
> Table B. Statistical comparison. Experiments follow the settings of Table 2 in the main paper, i.e., multi-device TTA. The accuracy is averaged over all devices and corruptions.
> |Method|Avg. Acc.|
> |-|-|
> |ETA|$60.5\pm1.8$|
> |**ETA+CoLA(Ours)**|$63.1\pm0.7$|
> |SAR|$58.6\pm0.1$|
> |**SAR+CoLA(Ours)**|$61.4\pm0.3$|
> |DeYO|$55.8\pm3.9$|
> |**DeYO+CoLA(Ours)**|$62.2\pm0.3$|
>
>
> > Q6. What are the overheads of the shared domain vectors? Can you provide any numerical example of the growth of shared vectors? How does the method achieve minimal computational overhead (only a 3-second increase) compared to the baseline ETA, as reported in Table 5?
>
> A. Compared to ETA, **CoLA only introduces an extra matrix ADD operation** in computational overhead to aggregate domain vectors before forward propagation. In this process, no extra forward passes is needed even with the increased domain vectors. During backpropagation, the previously learned domain vectors $\Delta\theta_i$ are frozen and CoLA only incurs extra update cost for the 1-d $\alpha$ to aggregate $\Delta\theta_i$ adaptively. **We do not alter the dimension of model weights or activations**, thus posing minimal computation and memory overhead in Table 5, i.e., additional 2.7% time cost and 0.08% memory footprint.
>
> Please also refer to Reviewer SsoW’s Q3-2, which demonstrates that CoLA can efficiently scale to 10,000 domain vectors with only an additional 11 seconds of latency.

---

> > ### Comment · Reviewer_L5Gu · 2024-08-08
> >
> > Thanks for the detailed response. The rebuttal addressed most of the concerns. Also, positioning the paper compared to Federated TTA and Cloud-Edge TTA makes the problem more novel.
> >
> > I would raise the scoring to 5.

---

> > > ### Author Response · Authors · 2024-08-08
> > > **Thank you for increasing your score!**
> > >
> > > Dear Reviewer L5Gu,
> > >
> > > Thank you for upgrading your score! Your comments are immensely beneficial in improving the quality of our paper.
> > >
> > > **We value your expertise and are happy to continue the discussion if you have further questions.**
> > >
> > > Best Regards,
> > >
> > > The Authors

---

### Official Review · Reviewer_SsoW · 2024-07-12

**Soundness:** 3
**Presentation:** 3
**Contribution:** 2
**Rating:** 5
**Confidence:** 4

**Summary:**

This paper explores test-time adaptation (TTA) in scenarios involving varying computational resources across multiple devices. To address this, the paper proposes storing and sharing domain knowledge among devices to improve adaptation. For resource-abundant devices (referred to as principal agents), domain knowledge is either newly learned or reprogrammed based on current inputs. Conversely, for resource-limited devices (referred to as follower agents), existing domain knowledge is efficiently aggregated according to domain similarities. The proposed method demonstrates superior performance across diverse experimental settings.

**Strengths:**

1. The proposed methods are technically sound.
2. The paper is well-organized and easy to understand.
3. Extensive analysis of various aspects and scenarios is provided, aiding in a better understanding of the proposed framework.

**Weaknesses:**

1. Consideration of Multi-Device Test-Time Adaptation Literature:

- While multi-device test-time adaptation literature is mentioned in the supplementary material, there are no direct mentions or experimental comparisons in the main paper. This omission may confuse readers about the paper's originality.
- The multi-device test-time adaptation literature should be more accurately described in the main paper, including conceptual comparisons in the introduction section.
- Benchmarking multi-device test-time adaptation methods (e.g., Cloud-Edge TTA [4], CEMA [5], FedTHE+ [21]) within the same experimental settings is crucial to demonstrate the superiority of the proposed method. The concepts of cloud and edge in [4,5] align with the notions of principal and follower agents in this paper.

2. Similarity to Prior Continual Test-Time Adaptation Work [A]:

This paper and prior continual test-time adaptation work [A] show high similarity in the following aspects:
- Both papers store and reuse domain knowledge for better adaptation.
- Both store domain knowledge in the affine parameters of norm layers.
- Both retrieve related domain knowledge based on the measured domain similarity between the current input and previous domains.
Given these similarities, it is essential to correctly reference the prior work [A] and include experimental comparisons on detailed design aspects.
[A] Test-time Adaptation in the Dynamic World with Compound Domain Knowledge Management, IEEE Robotics and Automation Letters, 2023

3. Additional Ablation Study and Analysis:

To improve the completeness of the paper, I suggest including the following:
- An ablation study on the impact of domain knowledge "stored from multi-devices" in multi-device scenarios. A baseline where each device has restricted access to its own domain knowledge. Comparing this baseline to the proposed method would highlight the impact of domain knowledge "stored from multi-devices."
- An analysis of the proposal's efficiency in terms of memory overload and inference time.
- An analysis of the ratio of selected domain knowledge for reprogramming/aggregation.
- Experimental results on the diverse order of domains for multiple devices.

4. Limited Representation of Real-World Scenarios:

While this paper targets more realistic scenarios, the experiments are somewhat limited in representing real-world conditions. It would be beneficial to include experimental results from (for example) autonomous driving scenarios under diverse weather conditions.

**Questions:**

1. Have you considered moving Figure A from the supplementary material to the main document? Figure A could help readers quickly grasp the overall concept of the proposed method.

2. Could the proposed methods be applied to the EATA method? Given that EATA is a strong baseline on several benchmarks, the synergy of the proposed method and EATA might set a new state-of-the-art.

**Limitations:**

This paper faithfully describes its limitations and potential negative impacts in general.

---

> ### Author Rebuttal · Authors · 2024-08-07
>
> > Q1. Comparison with Multi-Device Test-Time Adaptation.
>
> A. Please refer to our General Response.
>
> > Q2. Differences between CoLA and previous continual TTA work TTA-CDKM [A].
>
> A. The main differences between our CoLA and TTA-CDKM [A] are in the following aspects:
>
> 1. **Our CoLA is more feasible for adaptation on both computational resource-abundant and resource-limited devices than TTA-CDKM [A]**: TTA-CDKM [A] mainly focus on the adpdation on **one single resource-abundant device** in lifelong TTA scenarios. While our CoLA addresses the practical computation demands of both resource-abundant and resource-limited devices. More importantly, our CoLA facilitates a collaborative TTA community **across multiple devices with diverse resource budgets** rather than a single device.
> 2. **Our CoLA is more suitable for bandwith-limited senarios than TTA-CDKM [A]**. TTA-CDKM [A] updates the stored domain parameters when adapting to **each batch of test samples**. This introduces intensive communication cost for weight synchronization across devices post-adaptation. Conversely, our CoLA synchronizes the stored domain parameters only **when domain shift occurs**. It largely reduces the communication overhead to enable multi-device collaboration.
> 3. **Our CoLA makes better use of off-the-shelf parameters than TTA-CDKM [A]**. TTA-CDKM [A] assumes that all parameters are learned from domain adaptation and retrieves based on domain similarity, **failing to exploit off-the-shelf parameters without domain information**. In contrast, our CoLA adaptively learns the weighting $\alpha_i$ without relying on domain information. This enables our CoLA to **exploit more off-the-shelf parameters for rapid adaptation**, as shown in Table 7 in the main paper.
>
> We will discuss TTA-CDKM [A] in our revised paper. Due to the unavailability of TTA-CDKM [A] source code and the limited time, we did not include empirical comparisons with TTA-CDKM and leave this to the future.
>
>
> > Q3-1. An ablation study on the impact of domain knowledge "stored from multi-devices" in multi-device scenarios.
>
> A. We ablate the effects of knowledge "stored from multi-devices" in Table A, termed CoLA (NoShare). Here, CoLA (NoShare) only store the knowledge learned on its own device and do not share knowledge with other devices. As in this multi-device setting each device encounters fully different corruptions, SAR/ETA+CoLA (NoShare) achieves the similar performance as SAR/ETA. By leveraging knowledge from other devices, CoLA consistently boosts the performance of SAR and ETA, suggesting the importance of cross-device knowledge sharing.
>
> Table A. Ablation of cross-device knowledge sharing in CoLA. The experiments follow **the same settings as Table 2 in the main paper**. Due to page limits, we will include the *detailed results for each device with each corruption in the revision*.
>
> |Method|Avg. Acc.|
> |-|:-:|
> |NoAdapt|28.6|
> |SAR|58.6|
> |SAR+CoLA(NoShare)|58.6|
> |**SAR+CoLA(Ours)**|**61.7**|
> |ETA|61.3|
> |ETA+CoLA(NoShare)|61.3|
> |**ETA+CoLA(Ours)**|**63.7**|
>
> > Q3-2. An analysis of the proposal's efficiency in terms of memory overload and inference time.
>
> A. Our CoLA is both computation and memory efficient. In Table 5 of the main paper, we have provided latency and memory usage comparisons, in which both CoLA's follower and principal agents boost the performance a lot while maintaining nearly the same efficiency as NoAdapt/ETA baseline.
>
> In the response, we further illustrate CoLA’s efficiency with an increasing number of domain vectors. From Table B, CoLA efficiently scales to over 10,000 domain vectors, incurring only an additional 11s of runtime and 1,502MB of extra memory, yet remains substantially more efficient than CoTTA. We believe that 10,000 domain vectors should be adequate for handling most real-world applications with proper management.
>
>
>
> Table B. Efficiency comparison on ImageNet-C (Gaussian, level 5). The time is counted for processing 50,000 images on a single A100. $N$ is the number of domain vectors, which we initilize them as random weights in this table.
> |Method|ETA|+CoLA(N=1)|+CoLA(N=100)|+CoLA(N=10,000)|CoTTA|
> |-|:-:|:-:|:-:|:-:|:-:|
> |Time (s)|109|110|112|120|937|
> |Memory (MB)|7,433.2|7,433.7|7,448.2|8,935.5|21,628.6|
>
>
> > Q3-3. An analysis of the ratio of selected domain knowledge for reprogramming/aggregation.
>
> - **On Principal Agents (knowledge reprogramming):** When adapting to a new domain, CoLA favors knowledge that learns from a similar domain for rapid adaptation. However, as it captures more knowledge on the current domain, CoLA adaptively allocates more ratio to diverse domains for complementation (refer to Fig. I in our rebuttal PDF).
> - **On Follower Agents (knowledge aggregation):** With accurate domain similarity measurement, follower agents select relevant knowledge for seen domains. For unseen domains, CoLA assigns ratio more evenly while still prioritizing knowledge from a similar domain (refer to Fig. II in our rebuttal PDF).
>
>
>
>
> > Q3-4. Experimental results on the diverse order of domains for multiple devices.
>
> A. From Table I in our PDF file in Global Response, CoLA consistently outperforms baselines under 3 different domain orders, suggesting CoLA's stability.
>
> > Q4. Experiments under real-world scenarios.
>
> A. The datasets adopted by EATA effectively meet real-world demands. For instance, the ImageNet-A/V2/R datasets in Tables 6-7 are sourced from the real world. Additionally, the corruptions present in ImageNet-C are commonly encountered in real-world scenarios, such as when cameras degrade, weather changes, or cameras move. All these datasets are widely used in the TTA literature. Due to the time constraints of the rebuttal period, we will explore CoLA in more real-world scenarios in the future.
>
> > Q5. Could the proposed methods be applied to the EATA method?
>
> A. Yes, CoLA enhances the accuracy of EATA by 3.2% on ResNet-50. Please refer to Reviewer 1r2f's Q3 for more details.

---

> ### Author Response · Authors · 2024-08-10
> **Thanks and a kind reminder for discussion**
>
> Dear Reviewer SsoW,
>
> We would like to thank you for your invaluable feedback on improving the quality of our paper. In the previous response, we have re-clarified the novelty and contributions of our CoLA and demonstrated its efficiency and stability in diverse scenarios.
>
> We are wondering if our response has addressed your questions appropriately and can improve your opinion of our work? Kindly let us know if you might have further comments, and we will do our best to address them.
>
> Best Regards,
>
> The Authors

---

> ### Author Response · Authors · 2024-08-13
> **Thank you and looking forward to your response**
>
> Dear Reviewer SsoW,
>
> We appreciate your invaluable feedback and would be glad to include the new discussions and results in our revised paper.
>
>
> However, **if we have addressed your concerns, we kindly ask for your consideration in increasing the score for our paper.**
>
> We believe that our paper is distinct from existing literature and has made sufficient contributions, including:
>
> - To the best of our knowledge, **we are the first to study collaborative adaptation across multiple devices at test time and the first to enhance TTA on devices with varying resources**, meanwhile keeping privacy preserved and communication efficient.
> - **We devise two distinct collaborative learning paradigms for TTA.** With backpropagation, we jointly learn new domain-specific parameters and a reweighting term to reprogramm existing knowledge. With only forward passes, we aggregate different domain knowledge based on domain similarities in an optimization-free manner.
> - Our proposed CoLA **covers a wide range of real-world scenarios**, which include multi-devices with multi-domains (collaborative TTA), a single device with multi-domains (lifelong TTA), and a single device with a single domain (single-domain TTA).
> - We achieve notable results. Our proposed CoLA is plug-and-play with advanced TTA solutions and achieves **superior performance with negligible computation and memory overhead**. CoLA on forward-only agent significantly boosts the performance, *e.g.*, by 30% regarding accuracy on out-of-distribution samples, with nearly the same efficiency as standard inference.
>
>
> Our novelty and contribution are also recognized by other reviewers, such as:
>
> - *"The authors propose **a new and novel setting of cross-device collaborative TTA**."* [by **Reviewer L5Gu** and **Reviewer sr9b**]
> - *"The method **considers practical scenarios** such as communication cost, devices without training capability, etc."* [by **Reviwer sr9b** and **Reviewer 1r2f**]
> - *"The method is **plug-and-play with existing TTA methods**."* [by **Reviewer sr9b**]
> - *"CoLA is **simple and efficient**, demonstrating **excellent performance** in collaborative, lifelong, and single-domain TTA tasks."* [by **Reviewer 1r2f**]
>
> We sincerely hope our clarifications above can improve your opinion of our work and can help you reconsider your score. We look forward to your response and are happy to continue the discussion if you have further questions.
>
> Best Regards,
>
> The Authors

---

> > ### Comment · Reviewer_SsoW · 2024-08-13
> > **Response to the Author's rebuttal.**
> >
> > I have carefully reviewed the authors' responses and the comments from other reviewers. First of all, I would like to express my sincere appreciation for the authors' thoughtful and detailed responses. These have made me more positive about this submission. Below, I provide my thoughts on each of the authors' rebuttal points.
> >
> > Regarding Q1: The additional explanations provided in the general response have clarified the advantages of the proposed settings and methods. However, I still believe that the original submission did not fully emphasize these advantages, as the relevant explanations are scattered between the main paper and the supplementary material, and some crucial experimental comparisons are missing. Therefore, it is essential to include these new insights in a revised and reformatted manuscript.
> >
> > Regarding Q2: I appreciate the authors' careful discussion comparing their approach to TTA-CDKM, and I generally agree with their perspective. As the authors mentioned, including such a discussion would indeed highlight the superiority of the proposed methods.
> >
> > Regarding Q3-1: I have some questions regarding this part. According to the results, knowledge from the same devices does not seem to affect performance. I am curious about the nearly identical scores with and without CoLA (Noshare). Is there another reason that might explain this observation?
> >
> > Regarding Q3-2: Thank you for pointing out that the information on efficiency was already included. I apologize for overlooking this. The additional details provided are also very helpful.
> >
> > Regarding Q3-3, Q3-4, Q4, and Q5, I have also carefully read the authors' responses to these points. The additional analysis and results presented make sense. If this submission is accepted, I hope that this information will be properly included in the final version of the paper.
> >
> > In summary, could you please provide further clarification on the questions I raised regarding Q3-1? For the other points, I’ve shared my thoughts on various aspects of the paper. I believe it would be sufficient to share your perspective if there are any issues or misunderstandings in my comments.

---

> ### Author Response · Authors · 2024-08-14
> **Thanks for your feedback and further response**
>
> Dear Reviewer SsoW,
>
> We deeply appreciate your expertise and the detailed feedback. We provide further explanation regarding Q3-1 below.
>
> We would like to highlight that the success of CoLA mainly benefits from **reprogramming** previously-stored useful knowledge to aid current adaptation. Here, with and without CoLA(NoShare) achieves the same performance is because **Table A is a special case** in lifelong adaptation, in which reprogramming is not activated. In this special case, the same type of domains comes subsequently and occurs only once on each device, e.g., Gaussian Noise$\rightarrow$Shot Noise$\rightarrow$Impulse Noise. In this sense, reprogramming is unnecessary as the most useful knowledge for current adaptation **comes from exactly the previous domain**. Thus, CoLA has little room for improvement but, importantly, does not deteriorate performance.
>
>
> In addition to this special case, Baseline+CoLA(NoShare) can outperform Baseline in lifelong adaptation under various scenarios, such as when 1) **distinct domains come alternatively**, as evidenced in the first round of adaptation in Table 1, in which corruption order follows Gaussian Noise$\rightarrow$Snow$\rightarrow$Shot Noise; or when 2) **domains occur more than once**, e.g., Gaussian Noise$\rightarrow\cdots\rightarrow$Gaussian Noise. We further verify this in Table C by conducting long-term adaptation on each device following Table 2 of the main paper. Here, each device experiences two rounds of lifelong adaptation with identical domain orders. From Table C, CoLA enhances the accuracy of SAR by 2.2% at the second round of adaptation. This is because **reprogramming becomes critical to reuse useful knowledge from previous adaptations**.
>
>
> We believe that CoLA's flexibility to share and exploit learned knowledge would benefit TTA in more challenging scenarios and will explore more in our future works. We ensure to include the new discussions and results in our revised paper. Once again, we thank you for your thoughtful and invaluable reviews.
>
> Table C. Effectiveness of CoLA(NoShare) in lifelong TTA when domains occur more than once. We follow the settings as Table 2 in the main paper but extend it to the two-round lifelong adaptation for each device. For example, on Device 1, the domains come in the following order: N->B->W->D->N->B->W->D. We report average accuracy over three devices on each round of adaptation.
>
> |Method|Round 1|Round 2|
> |-|:-:|:-:|
> |SAR|58.6|59.8|
> |SAR+CoLA(NoShare)|58.6|62.0|

---

> > ### Comment · Reviewer_SsoW · 2024-08-14
> > **Thanks for author's feedback**
> >
> > Thank you for your detailed feedback regarding Q3-1.
> >
> > Overall, the authors' responses have thoroughly addressed my original concerns. While the original manuscript missed some important aspects, I will increase my score toward the positive side, trusting that the authors will make careful revisions. Thank you again for your comprehensive feedback.

---

> > > ### Author Response · Authors · 2024-08-14
> > > **Thank you for increasing your score!**
> > >
> > > Dear Reviewer SsoW,
> > >
> > > Thank you for upgrading your score!
> > >
> > > Your comments are immensely beneficial in improving the quality of our paper. We will be glad to include the new discussions and results in our revised paper.
> > >
> > > Best Regards,
> > >
> > > The Authors

---

### Official Review · Reviewer_1r2f · 2024-07-13

**Soundness:** 3
**Presentation:** 4
**Contribution:** 3
**Rating:** 7
**Confidence:** 5

**Summary:**

This paper considers test-time adaptation in multi-device scenarios and proposes Collaborative Lifelong Adaptation (CoLA). Domain knowledge vectors, as interactive objects on devices, are maintained and stored by measuring KL-divergence between the current test batch and the online estimated distribution. This paper categorizes different devices into principal agents and follower agents based on computational resources and latency demands, and uses different strategies for online prediction or parameter updates during testing. Experiments in various TTA settings demonstrate that CoLA has good adaptability and anti-forgetting capabilities. Rigorous ablation studies well support the viewpoints in the paper.

**Strengths:**

CoLA is simple and efficient, demonstrating excellent performance in collaborative, lifelong, and single-domain TTA tasks. It also considers the application for devices with different computational resources and latency demands in practice. This paper is well-organized and easy to follow.

**Weaknesses:**

- The hyperparameter z is used for detecting distribution shift and influences the storage of new domain vectors. The paper lacks ablation studies on the threshold z.
- In the setting of lifelong TTA, the smooth and stable updating of the affine parameters of norm layers relies on a sufficiently large batch size. It is necessary to analyze the robustness of CoLA under conditions where the batch size is extremely small.
- It’s insufficient to use only ViT-Base as the source model in the main experiments. Analyzing the effects of CoLA on different source models can provide more comprehensive evidence, such as ResNet and MobileNet.
- (minor) The experiments are all targeted at classification. It is intriguing to explore the effectiveness of CoLA in detection and segmentation tasks.

**Questions:**

It appears that domain knowledge vectors play a pivotal role in CoLA. Apart from using the affine parameters of norm layers, are there any other potential methods to determine these vectors?  (Just in terms of discussion)

I hope that the code can be released.

**Limitations:**

Yes

---

> ### Author Rebuttal · Authors · 2024-08-07
>
> > Q1. Ablation studies on the threshold z.
>
> A. CoLA remains effective among a wide range of threshold z, as shown in Table A. From the results, while a stricter threshold saves more domain vectors, CoLA achieves a stable performance of around 64.7%. When threshold z increases, CoLA saves significantly fewer domain vectors and still enhances the performance significantly, *i.e.*, the average accuracy of 62.2\% in ETA+CoLA (z=10) *vs.* 46.4% in ETA.
>
>
>
> Table A. Sensitivity of threshold z. Experiments follow the settings of Table 1 in the main paper, i.e., single-device lifelong adaptation, and CoLA is incorparated with ETA. We report average accuracy over 10 rounds with each round consists of 15 corruptions of ImageNet-C. The average accuracy of ETA baseline is 46.4%.
> ||CoLA, z=0.01|CoLA, z=0.05|CoLA, z=0.1|CoLA, z=1|CoLA, z=10|
> |-|:-:|:-:|:-:|:-:|:-:|
> |Avg Acc.|64.7|64.6|64.8|63.8|62.2|
> |#Saved domain vectors|20740|369|169|110|48|
>
>
> > Q2. The robustness of CoLA under small batch sizes.
>
>
> A. The stability of CoLA under small batch sizes is primarily determined by the base algorithms, such as ETA and SAR, rather than CoLA itself. This is because CoLA is a plug-and-play module designed to be incorporated with existing methods. In the response, we provide further empirical results to verify CoLA under small batch sizes. From Table B, SAR+CoLA achieves consistent results across different batch sizes, without performance degradation as the batch size reduces from 16 to 2. These results show that CoLA is less sensitive to small batch sizes.
>
>
> Table B. Performance of CoLA under various batch sizes. We follow the same settings of Table 2 in the main paper and report **average accuracy** over all devices and corruptions here.
> |Method|BS=16|BS=4|BS=2|
> |-|-|-|-|
> |SAR|58.8|58.9|58.7|
> |**SAR+CoLA**|61.6|61.7|61.7|
> |ETA|60.5|58.5|57.1|
> |**ETA+CoLA**|62.8|61.7|61.5|
>
> > Q3. Analyzing the effects of CoLA on different source models, such as ResNet.
>
> A. Table C demonstrates the effectiveness of our CoLA on ResNet-50, where CoLA updates and stores the affine parameters of batch normalization layers in ResNet.
>
> From Table C, CoLA consistently enhances the performance of ETA/EATA/SAR throughout 10 rounds of adaptation and addresses the issue of performance degradation in long-term adaptation. These results are consistent with Table 1 in the main paper using ViTBase, further indicating CoLA's effectiveness in accumulating and exploiting learned knowledge for lifelong adaptation.
>
> Table C.  Effectiveness of CoLA on ResNet in the lifelong TTA scenarios. We follow the same settings as Table 1 in the main paper.
> |Round|1|4|7|10|Avg (over 10 rounds)|
> |-|:-:|:-:|:-:|:-:|:-:|
> |NoAdapt|18.0|18.0|18.0|18.0|18.0|
> |CoTTA|33.4|2.2|1.2|1.2|7.6|
> |ETA|42.4|37.6|35.9|34.7|37.3|
> |**ETA+CoLA**|46.5|49.5|49.9|49.8|49.3|
> |EATA|47.5|46.7|46.4|46.2|46.7|
> |**EATA+CoLA**|48.2|50.1|50.3|50.2|49.9|
> |SAR|35.9|36.2|35.8|36.1|24.4|
> |**SAR+CoLA**|39.4|43.5|44.7|45.3|43.6|
>
>
>
> > Q4. The experiments are all targeted at classification. It is intriguing to explore the effectiveness of CoLA in detection and segmentation tasks.
>
>
> A. Thank you for your valuable suggestion. We fully agree that applying CoLA to more tasks is very intriguing. However, due to the time limits of the rebuttal period, we plan to further explore this in the future.
>
>
> > Q5. Apart from using the affine parameters of norm layers, are there any other potential methods to determine these vectors? (Just in terms of discussion)
>
> A. Thanks for your interesting question. In addition to affine parameters of norm layers, CoLA can utilize any parameter-efficient fine-tuning methods to select learnable parameters as the knowledge vectors to maintain high memory efficiency. This includes LoRA-based methods and Prompt Tuning methods. Table 7 in our main paper also verifies the effectiveness of CoLA with prompt tuning, enhancing TPT's accuracy by 1.5% on ImageNet.
>
>
> > Q6. I hope the code can be released.
>
> A. Yes, we will release our code as soon as the paper is accepted.

---

> > ### Comment · Reviewer_1r2f · 2024-08-13
> >
> > Thanks for the response.
> >
> > The rebuttal addressed all my concerns.
> >
> > I will keep my score.

---

> > > ### Author Response · Authors · 2024-08-13
> > > **Thanks again for your invaluable feedback!**
> > >
> > > Dear Reviewer 1r2f,
> > >
> > > We are glad to hear that all your concerns are addressed. Thanks again for your expertise and invaluable feedback on improving the quality of our paper!
> > >
> > > Best Regards,
> > >
> > > The Authors

---

> ### Author Response · Authors · 2024-08-10
> **Thanks and a kind reminder for discussion**
>
> Dear Reviewer 1r2f,
>
> We would like to thank you for recognizing the contributions of our work and also for the invaluable feedback. We are wondering if our response has addressed your questions appropriately and can further improve your opinion of our work.
>
> We truly value your expertise and are happy to continue the discussion if you have further questions.
>
> Best Regards,
>
> The Authors

---

### Official Review · Reviewer_sr9b · 2024-07-13

**Soundness:** 3
**Presentation:** 3
**Contribution:** 3
**Rating:** 5
**Confidence:** 5

**Summary:**

Prior test-time adaptation (TTA) methods focused on single-device setups only, despite models being deployed to multiple devices in the real world. This paper presents Collaborative Lifelong Adaptation (CoLA) to leverage this situation and improve performance. Specifically, this paper introduces Principal Agents and Follower Agents concepts and proposes knowledge reprogramming, similarity-based knowledge aggregation, and automatic domain shift detection for efficient collaboration. Experiments with various settings and state-of-the-art (SOTA) TTA methods demonstrate the effectiveness and applicability of CoLA.

**Strengths:**

1. Interesting problem of utilizing knowledge from multiple devices for collaborative TTA.
2. The method considers practical scenarios such as communication cost, devices without training capability, etc.
3. The method is plug-and-play with existing TTA methods.

**Weaknesses:**

1. Lack of deeper analysis on when prior knowledge helps and when it harms. As environments are diverse and KL-divergence-based distance metrics are not always accurate in neural networks, utilizing knowledge from other devices might be harmful. It would be better to have a deeper investigation into when prior knowledge is beneficial, how much prior knowledge is required to get the benefit, the extent of the benefit, and how many such cases are observed in the experiments.
2. The domain shift setup is a bit unrealistic. In real-world scenarios, domain shifts are often gradual rather than sudden. It is unclear whether the proposed method can handle gradual domain shift scenarios.
3. No error bars are reported, which weakens the statistical significance of the results.
4. This submission doesn't include code, which limits the validity and transparency of the work.

**Questions:**

1. What is the impact of this initialization strategy compared to randomized/equal weights?
2. Where does the communication cost reduction come from? Is it only from utilizing BN layers?
3. What is the accuracy of automatic domain detection?
4. Why is CoLA better than the baseline in the first round as well, before the shift happens (Table 1)?

**Limitations:**

The authors adequately addressed the limitations.

---

> ### Author Rebuttal · Authors · 2024-08-07
>
> > Q1. Analyses on when prior knowledge helps and when it harms.
>
> A. **Analyses on when prior knowledge helps**: In our paper, we seek to improve test-time model adaptation performance and efficiency by exploiting useful knowledge from multiple devices in the application environment. In practice, different devices may frequently encounter similar (e.g., Gaussian blur and Motion blur) or even identical testing domains. For these cases, our CoLA paradigm can reprogram these similar/useful knowledge via learnable re-weighting term when similar domain data are encountered. This enables CoLA to adapt quickly while simultaneously enhancing adaptation performance.
>
> In fact, prior knowledge does not help only if all knowledge/domain vectors learned from previously encourtered domains perform no better than the source pretrained model, which is however unrealistic given the similarities in real-world environments.
>
> **Analyses on when prior knowledge harms**: Note that our paradigm conducts model adaptation using prior knowledge to minimize the test-time objective. **If the prior knowledge of some devices is harmful, using such knowledge shall not decrease the test objective.** Thus, such prior knowledge will not be used for model adaptation. In this sense, our method is stable even with harmful prior knowledge (*e.g.*, the domain knowledge in some devices are not helpful for principal agents to do TTA). Our stability is also benefited from our initialization, please refer to Q5 for more discussions.
>
>
> > Q2. CoLA's effectiveness under gradual domain shift (GDS) scenarios.
>
> A. CoLA works well under GDS scenarios as shown in Table C. Here, we follow the benchmark [A,B] to evaluate our CoLA under GDS, in which the severity of each corruption gradually changes during a lifelong adaptation process. From the results, CoLA effectively enhances the accuracy of ETA by 1.8%.
>
> Table B. Accuracy of CoLA on ImageNet-C under gradual domain shifts [A,B].
> |NoAdapt|ETA|+ CoLA(Ours)|
> |:-:|:-:|:-:|
> |53.2|71.2|73.0|
>
> [A] Gradual test-time adaptation by self-training and style transfer
>
> [B] Robust Mean Teacher for Continual and Gradual Test-Time Adaptation
>
>
> > Q3. No error bars are reported.
>
> A. Please refer to Q5 for Reviewer L5Gu.
>
> > Q4. Code Release.
>
> A. We will release the source code upon acceptance.
>
> > Q5. What is the impact of this initialization strategy compared to randomized/equal weights?
>
> A. Our initialization strategy helps stabilize TTA, particularly when the shared domain vectors contain harmful prior knowledge. Randomly or equally initializing $\alpha_i$ may select harmful knowledge that deteriorates model performance, as in Table C. In contrast, our strategy achieves consistent adaptation performance even when the proportion of harmful parameters exceeds 96%, demonstrating its superiority. Additionally, this strategy also encourages fast adaptation in the early phase of new domain data arrival.
>
> Table C. Effects of different initialization strategies w.r.t. $\alpha$. CoLA uses only N harmful prior knowledge (random weights). Results (Accuracy) obtained on ImageNet-C (Gaussian, level 5) with ETA+CoLA.
> ||N=0|N=2|N=4|N=100|N=1000|
> |-|:-:|:-:|:-:|:-:|:-:|
> |Ours|51.97|52.02|52.04|52.04|52.04|
> |Random $\alpha$|51.97|10.46|0.15|0.1|0.1|
> |Equal  $\alpha$|51.97|0.1|0.1|0.1|0.1|
>
> > Q6. Where does the communication cost reduction come from? Is it only from utilizing BN layers?
>
> A. The reduction comes from both **transmitting Norm Layers** and **transmitting only when a domain shift occurs**. For example, using ViT and ImageNet-C data, CoLA requires only 0.15MB per domain (with accurate domain shift detection). In contrast, the previous method, CEMA, transmits both raw data and norm layers for each iteration, resulting in a communication cost of about 568.85MB for a single domain.
>
> > Q7. What is the accuracy of automatic domain detection?
>
> A. Our detector achieves 100% domain detection accuracy in Table 1 of the main paper (single-device lifelong adaptation) and 71.4% in Table 2 of the main paper (multi-device TTA). However, we emphasize that the lower detection accuracy in Table 2 does not compromise CoLA's performance. As in Table D, CoLA achieves the comparable final accuracy as CoLA with an oracle domain detector. This is because certain domains in Table 2 share similarities. Therefore, it is not necessary to identify and save all domain vectors accurately, which helps reduce redundant domain vector storage.
>
>
> Table D. Avg accuracy comparison with CoLA+[Oracle domain shift detector]. Experiments follow the settings of Table 2 in main paper.
> |ETA|ETA+CoLA(Ours)|ETA+CoLA(Oracle)|
> |:-:|:-:|:-:|
> |61.3|63.7|63.7|
>
>
> > Q8. Why is CoLA better than the baseline in the first round as well, before the shift happens (Table 1)?
>
> A. This is because each round consists of 15 different corruptions, and each element in Table 1 represents their average result. The 15 corruptions fall into four main types: noise, blur, weather, and digital. Within each category, the corruptions share certain similarities. Thus, for example, adaptation to Impulse Noise can benefit from knowledge learned on Gaussian Noise. CoLA leverages this similar knowledge to aid subsequent adaptation, thus achieving better performance.

---

> > ### Comment · Reviewer_sr9b · 2024-08-08
> > **Thank you for the reseponse**
> >
> > Thanks for the rebuttal. I've read it carefully and it addressed most of my concerns. I will keep my score on the positive side.

---

> ### Author Response · Authors · 2024-08-09
> **Thank you for your positive feedback!**
>
> Dear Reviewer sr9b,
>
> We are happy to hear that your concerns have been addressed and sincerely thank you for recognizing our novelty and practicality.
>
> In this sense, **could we kindly request you to reconsider the possibility of improving your initial score?**
>
> **We truly value your expertise and are hopeful for a favorable reconsideration.** We are happy to continue the discussion if you have further questions.
>
> Best Regards,
>
> The Authors

---

### Author Rebuttal · Authors · 2024-08-07

We deeply appreciate all reviewers for their valuable feedbacks and constructive comments on improving the quality of our paper. We would like to address your questions below.

>  G1. Differences and advantages of our methods from/over federated TTA [A,B]:

- **Differences on problem settings and learning paradigms:** Federated TTA [A,B] focuses on addressing data silos/privacy issues by exploiting collaborative learning during the federated source training phase **(supervised)**. However, it performs TTA on each device independently during the testing phase. In our paper, we seek to improve test-time model adaptation performance and efficiency by exploiting knowledge from multiple devices in the application environment, which essentially is a new **unsupervised** on-time TTA paradigm.
- **Differences on application flexibilities in deployment:** In federated TTA [A,B], the training phase and test-time adaptation phase are highly correlated, which means it can only use its own trained model for TTA. This makes federated TTA [A,B] restricted for real-world applications. In contrast, our CoLA paradigm can be applied to any pre-trained models, given that feedbacks from multiple devices or single device are available. Thus, it offers much better flexibility in deployment.
- **Differences on TTA paradigms in the testing phase**. In federated TTA [A,B], each device conducts model adaptation using normal TTA strategies independently, which means it inherits the limitations of the used TTA methods. In contrast, our method is able to exploit knowledge from multiple devices to improve both the performance and efficiency of TTA.
- In fact, **our CoLA can be incorporated into federated TTA with slight efforts to boost its performance.** As shown in Table A, the performance of the federated TTA[A] can be much boosted with our CoLA learning paradigm.



Table A. Performance improvement of  Federated TTA [A] with our CoLA on CIFAR-10-C with ResNet-18. **The source model is trained by federated TTA [A].**
|Method|NoAdapt|Federated TTA [A]|Federated TTA [A]+CoLA(Ours)|
|:-:|:-:|:-:|:-:|
|Acc.|65.2|71.2|**74.3**|


[A] Adaptive test-time personalization for federated learning. NeurIPS 2023.

[B] Test-time robust personalization for federated learning. ICLR 2023.


> G2. Differences and advantages from/over Cloud-Edge TTA (CEMA) [C]:

- **Differences on problem settings**: CEMA [C] is dedicated to cloud-edge collaborative TTA between two devices, *i.e.*, one cloud server and one edge device. While our CoLA focuses on **collaborative TTA** among multiple devices with diverse computational resource budgets.
- **Differences on communication overhead**: CEMA [C] transmits raw samples and model parameters for each learning iteration/batch, leading to relatively high communication overhead. In contrast, our CoLA only transmits domain vectors (affine parameters of norm layers) when a domain shift is detected, resulting in **much lower communication costs**. Taking ViT-Base and ImageNet-C as an example, the communication cost for a single domain is 0.15MB (Ours) vs. 568.85MB (CEMA).
- **Differences on privacy protection**: CEMA [C] directly uploads raw samples from the edge devices to the cloud, and thus suffers privacy concerns. Instead of transmitting raw samples, our CoLA only transmits domain vectors/norm layer parameters, enhancing its availability for privacy-sensitive applications.
- Moreover, our CoLA achieves better adaptation performance than CEMA [C]. In Table C, our CoLA equipped with SAR attains a higher accuracy than CEMA (61.7% *vs.* 59.4%).

**We will include discussions with methods [A,B,C] in the revised paper.**

Table C. Comparison between CEMA and SAR+CoLA(Ours) on ImageNet-C (level 5). The experiments follow the settings of Table 2 in the main paper. Here, we report only the average results across all devices and corruptions.
| Method             | Avg. Acc. |
| ------------------ | :-------: |
| NoAdapt            |   28.6    |
| Cloud-Edge TTA (CEMA) [C]            |   59.4    |
| SAR                |   58.6    |
| **SAR+CoLA(Ours)** | **61.7**  |


[C] Towards robust and efficient cloud-edge model adaptation via selective entropy distillation. ICLR 2024.

---

### Decision · Program_Chairs · 2024-09-25

**Decision:**

Accept (poster)

**Comment:**

This paper considers test-time adaptation in multi-device scenarios and proposes Collaborative Lifelong Adaptation (CoLA). Domain knowledge vectors, as interactive objects on devices, are maintained and stored by measuring KL-divergence between the current test batch and the online estimated distribution. This paper categorizes different devices into principal agents and follower agents based on computational resources and latency demands, and uses different strategies for online prediction or parameter updates during testing.

The reviewers were initially mixed with both positive and negative scores. They agreed that the approach is novel and interesting, the paper is well written, the method is of practical importance. The positive reviewers noted that CoLA is simple and efficient, demonstrating good performance in collaborative, lifelong, and single-domain TTA tasks. It also considers the application for devices with different computational resources and latency demands in practice, and  is technically sound. The negative reviewers mostly raised concerns about lack of citations and comparisons to previous work, limited experimental evidence, and various other questions of detail. The authors addresses these concerns in their rebuttal and extensive discussion with the reviewers. In the end, all reviewers were positive towards the paper. The ACs concur that the paper merits publications